



**1. Title page:**
**Complex controls on nitrous oxide flux across a long elevation gradient in the tropical**
**Peruvian Andes**
Torsten Diem[1,2], Nicholas J. Morley[1], Adan Julian Ccahuana[3], Lidia Priscila Huaraca Quispe[3],
Elizabeth M. Baggs[4], Patrick Meir[5,6], Mark I.A. Richards[1], Pete Smith[1], and Yit Arn Teh[1,2]*
[1] School of Biological Sciences, University of Aberdeen, UK
[2] Formerly at the School of Geography and Geosciences, University of St Andrews, UK
[3] Universidad Nacional de San Antonio Abad del Cusco, Peru
[4] The Royal (Dick) School of Veterinary Studies, University of Edinburgh
[5] School of GeoSciences, University of Edinburgh, UK
[6] Research School of Biology, Australian National University, Canberra, Australia
* Corresponding author; yateh@abdn.ac.uk



## 2. Abstract

Current bottom-up process models suggest that montane tropical ecosystems are weak atmospheric sources of $N_2O$, although recent empirical studies from the southern Peruvian Andes have challenged this idea. Here we report $N_2O$ flux from combined field and laboratory experiments that investigated the process-based controls on $N_2O$ flux from montane ecosystems across a long elevation gradient (600-3700 m a.s.l.) in the southern Peruvian Andes. Nitrous oxide flux and environmental variables were quantified in four major habitat types (premontane forest, lower montane forest, upper montane forest and montane grassland) at monthly intervals over a 30-month period from January 2011 to June 2013. The role of soil moisture content in regulating $N_2O$ flux was investigated through a manipulative, laboratory-based $^{15}N$-tracer experiment. The role of substrate availability (labile organic matter, $NO_3^-$) in regulating $N_2O$ flux was examined through a field-based litter-fall manipulation experiment and a laboratory-based $^{15}N$-$NO_3^-$ addition study. Ecosystems in this region were net atmospheric sources of $N_2O$, emitting $0.27 \pm 0.07$ mg $N$-$N_2O$ $m^{-2}$ $d^{-1}$. Nitrous oxide flux was inversely related to elevation; $N_2O$ flux was greatest in premontane forest ($0.75 \pm 0.18$ mg $N$-$N_2O$ $m^{-2}$ $d^{-1}$), followed by lower montane forest ($0.46 \pm 0.24$ mg $N$-$N_2O$ $m^{-2}$ $d^{-1}$), montane grasslands ($0.07 \pm 0.08$ mg $N$-$N_2O$ $m^{-2}$ $d^{-1}$), and upper montane forest ($0.04 \pm 0.07$ mg $N$-$N_2O$ $m^{-2}$ $d^{-1}$). Nitrous oxide flux showed weak seasonal variation across the region; only lower montane forest showed significantly higher $N_2O$ flux during the dry season compared to wet season. Manipulation of soil moisture content in the laboratory indicated that $N_2O$ flux was significantly influenced by changes in water-filled pore space (WFPS). The relationship between $N_2O$ flux and WFPS was bimodal and non-linear, diverging from theoretical predictions of how WFPS relates to $N_2O$ flux. Nitrous oxide flux was greatest at 90 and 50 % WFPS, and lowest at 70 and 30 % WFPS. This bimodal distribution of $N_2O$ flux suggests a complex relationship between WFPS, environmental variables, and nitrate-reducing processes. Changes in labile organic matter inputs, through the manipulation of leaf litter-fall, did not alter $N_2O$ flux, suggesting that litter inputs have a negligible impact on $N_2O$ flux. Nitrate addition experiments demonstrated that variations in $NO_3^-$ availability constrained $N_2O$ flux. Habitat – a proxy for $NO_3^-$ availability under field conditions – was the best predictor for $N_2O$ flux, with N-rich habitats (premontane forest, lower montane forest) showing significantly higher $N_2O$ flux than N-poor habitats (upper montane forest, montane grassland). Nitrous oxide flux did not respond to short-term changes in $NO_3^-$ concentration.







**3. Introduction**
The tropics are the largest source of atmospheric nitrous oxide ($N_2O$), accounting for at least
half of all global emissions (Hirsch et al., 2006;Huang et al., 2008;Kort et al., 2011;Nevison et
al., 2007;Saikawa et al., 2014). The bulk of tropical $N_2O$ emissions come from terrestrial
sources, with the largest emissions arising from agricultural land and unmanaged lowland
tropical forests (Hirsch et al., 2006;Huang et al., 2008;Kort et al., 2011;Nevison et al.,
2007;Saikawa et al., 2014). However, while we have a relatively robust understanding of the
global atmospheric budget as a whole (Hirsch et al., 2006;Huang et al., 2008;Saikawa et al.,
2014), our knowledge of regional atmospheric budgets, particularly at the sub-continental
scale, is much more limited, due to the constraints imposed by the spatial distribution of
existing atmospheric sampling networks and ground-based, ecosystem-scale sampling
efforts (Kort et al., 2011;Nevison et al., 2004;Nevison et al., 2007;Saikawa et al., 2014).

In order to predict and model $N_2O$ flux at these smaller (sub-continental) spatial scales,
bottom-up emissions inventories or process-based models are often used, with emissions
estimates constrained by empirical measurements (Werner et al., 2007;Li et al., 2000;Potter
et al., 1996;Saikawa et al., 2013). However, these models are only as reliable as the data
used to parameterize them; as a consequence, ecosystems that are under-represented in
the empirical literature or which are poorly understood may be modelled less accurately,
with knock-on effects for larger-scale emissions estimates (Saikawa et al., 2013;Teh et al.,
2014;Werner et al., 2007). Nitrous oxide dynamics in montane tropical ecosystems are
particularly poorly understood, because past research has concentrated on $N_2O$ flux from
lowland *tierra firme* forests (Saikawa et al., 2013;Teh et al., 2014;Werner et al., 2007).
Montane ecosystems, however, are important components of many tropical landscapes, and
account for a sizeable land area. For example, in continental South America, montane
ecosystems (>500 m a.s.l.) cover more than 8 % of the land surface (Eva et al., 2004), and
play key roles in regional carbon (C), nitrogen (N), and greenhouse gas (GHG) dynamics
(Girardin et al., 2010;Moser et al., 2011;Teh et al., 2014;Wolf et al., 2012;Wolf et al., 2011).
Process-based models predict that $N_2O$ flux from these montane environments are lower
than those from the lowland tropics (i.e. <1.0 kg $N_2O$-N $ha^{-1}$ $yr^{-1}$) (Saikawa et al.,

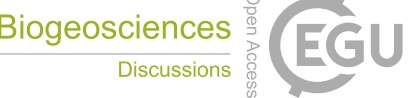

2013;Werner et al., 2007). However, these models have rarely been tested against empirical
data, and several field studies indicate that $N_2O$ flux from montane ecosystems can exceed
these prior models' estimates (Corre et al., 2010;Teh et al., 2014;Veldkamp et al., 2008). In
some instances, $N_2O$ flux from montane ecosystems can in fact approach emissions from
lowland forests, begging the question as to whether or not existing models do, in fact,
accurately represent flux from these high elevation ecosystems (Corre et al., 2010;Teh et al.,
2014;Veldkamp et al., 2008).

In order to improve our wider understanding of the dynamics and biogeochemistry of $N_2O$ in
montane tropical forests, we conducted a combination of field- and laboratory-based studies
to investigate the environmental controls on denitrification and $N_2O$ flux across a long
elevation gradient (600-3700 m a.s.l.) in the tropical Peruvian Andes. Prior work from this
region indicated that montane ecosystems in this region were stronger sources of $N_2O$ than
predicted by prior bottom-up process models (Teh et al., 2014). In particular, lower elevation
premontane and lower montane forests, which are areally-dominant in this region, showed
emission rates that are on par with lowland tropical forests, suggesting that these
ecosystems could be important contributors to regional atmospheric budgets (Teh et al.,
2014). Nitrous oxide flux appeared to be derived from  (i.e. denitrification, dissimilatory  to
ammonium), and were linked to seasonal variations in climate, with $N_2O$ emissions
increasing during the dry season compared to the wet season (Teh et al., 2014). However,
contrary to theoretical expectations (Davidson, 1991;Firestone and Davidson,
1989;Groffman et al., 2009), $N_2O$ flux was not directly influenced by soil moisture content in
our field dataset (Teh et al., 2014), raising important questions about the role of soil
moisture as a proximate driver of $N_2O$ flux. Nitrous oxide flux appeared to be more strongly
constrained by the availability of substrates for , particularly the availability of nitrate ($NO_3^-$)
(Teh et al., 2014).

In this study, we extended our time series to multi-annual time scales, in order to better
understand the role of longer-term climatic variability in modulating $N_2O$ flux, and to
investigate the mechanistic controls on $N_2O$ flux (e.g. substrate availability, soil moisture) in
greater detail. We also conducted a series of complementary field and laboratory
experiments to evaluate key process-based controls on $N_2O$ flux, such as soil moisture




content, labile carbon availability, and $NO_3^-$ availability. The overarching goals of this
research were to: investigate how climate and environmental variables regulate $N_2O$ flux
over multi-annual time scale; clarify the role of soil moisture as a proximate or distal driver
of $N_2O$ flux; and evaluate the role of key substrates, such as labile organic matter and $NO_3^-$,
for driving $N_2O$ flux. Specifically, we hypothesized that:

118       **H1.** *Seasonal variations in key environmental variables (e.g. soil moisture content, $NO_3^-$)*

119           *drive patterns in $N_2O$ flux on multi-annual time scales*

120       **H2.** *$N_2O$ flux increases proportionately with soil moisture content*

121       **H3.** *$N_2O$ flux increases proportionately with the availability of substrates for nitrate*

122           *reduction (i.e. labile organic matter, $NO_3^-$)*

To address these hypotheses, we conducted a combined field and laboratory study,
including monthly field flux measurements collected across a range of elevations and
habitats over a 30-month period; a laboratory-based soil moisture manipulation experiment;
a field-based litter-fall manipulation study; and a laboratory-based $NO_3^-$ addition study.


**4. Materials and methods**
**4.1 Study site**
Measurements were conducted on the eastern slope of the Andes in the Kosñipata Valley,
Manu National Park, Peru (Figure 1) (Malhi et al., 2010). This $3.02 \times 10^6$ ha (30,200 km$^2$)
region has been the subject of intensive ecological, biogeochemical and climatological
studies since 2003 by the Andes Biodiversity and Ecosystem Research Group (or, ABERG;
http://www.andesconservation.org), and contains a series of long-term permanent plots
across a 200-3700 m above sea level (m a.s.l) elevation gradient that stretches from the
western Amazon to the Andes (Malhi et al., 2010). This part of the Andes experiences
pronounced seasonality in rainfall but not in air temperature; the dry season extends from
May to September and the wet season from October to April (Girardin et al., 2010). Thirteen
sampling plots (approximately 20 x 20 m each) were established at four different habitats
across a gradient spanning 600-3700 m a.s.l., including premontane forest (600 – 1200 m
a.s.l.; n = 3 plots), lower montane forest (1200 – 2200 m a.s.l.; n = 3 plots), upper montane
forest (2200 – 3200 m a.s.l.; n = 3 plots), and montane grasslands (3200 – 3700 m a.s.l.; n = 4
plots; colloquially referred to as "puna") (Figure 1). In premontane forest, sampling plots





were established in Hacienda Villa Carmen, a 3,065 ha biological reserve operated by the
Amazon Conservation Association (ACA), containing a mixture of old-growth forest,
secondary forest and agricultural plots (Teh et al., 2014). Sampling for soil gas flux was
concentrated in the old-growth portions of the reserve. For lower montane and upper
montane forests, sampling plots were established adjacent to or within existing 1 ha
permanent sampling plots established by ABERG (Teh et al., 2014). Sampling plots were also
established in montane grasslands (Teh et al., 2014). To capture a representative range of
environmental conditions, mesotope-scale (100 m-1 km scale landforms) topographic
features were sampled (Belyea and Baird, 2006). Mesotopic features include ridges, slopes,
flats and a high elevation basin. The latter two landforms include wet, grassy lawns with no
discernible grade, and a peat-filled depression, respectively. Summary site descriptions are
provided in Table 1. Data on soil properties were collected as part of this study, while mean
annual precipitation is from earlier research by ABERG (Girardin et al., 2010).

**4.2 Soil-atmosphere exchange**
Field sampling was performed over a 30-month period from January 2011 to June 2013 for
all habitats except for premontane forest. Because of circumstances outside our control,
only 24-months of data were collected for premontane forest, with sampling commencing in
July 2011. Soil-atmosphere flux was collected monthly, except where flooding or landslides
prevented safe access by investigators to the study sites. Gas exchange rates were
determined with five replicate gas flux chambers deployed in each of the thirteen plots (n =
65 flux observations per month). All representative landforms were sampled in each habitat
(Table 1).

Soil-atmosphere flux of $CH_4$, $N_2O$ and $CO_2$ were determined using a static flux chamber
approach (Livingston and Hutchinson, 1995), although only $N_2O$ flux are reported here.
Methane and $CO_2$ flux are discussed in detail in another publication (Jones et al., 2016).
Static flux chamber measurements were made by enclosing a 0.03 $m^2$ area with cylindrical,
opaque (i.e. dark), two-component (i.e. base and lid) vented chambers. Chamber bases were
permanently installed to a depth of approximately 5 cm and inserted >1 month prior to the
commencement of sampling, in order to minimise potential artefacts from root mortality
following base emplacement (Varner et al., 2003). Chamber lids were fitted with small



computer case fans to promote even mixing in the chamber headspace (Pumpanen et al.,
2004). Headspace samples were collected from each flux chamber over a 30-minute
enclosure period, with samples collected at 4 discrete intervals using a gastight syringe. Gas
samples were stored in evacuated Exetainers® (Labco Ltd., Lampeter, UK), shipped to the UK
by courier, and subsequently analysed for $CH_4$, $N_2O$ and $CO_2$ concentrations with a Thermo
TRACE GC Ultra (Thermo Fisher Scientific Inc., Waltham, Massachusetts, USA) at the
University of St Andrews. Chromatographic separation was achieved using a Porapak-Q
column, and analyte concentrations quantified using a flame ionization detector (FID) for
$CH_4$, electron capture detector (ECD) for $N_2O$, and methanizer-FID for $CO_2$. Instrumental
precision was determined by repeated analysis of standards and was better than 5 % for all
detectors. Gas flux rates were determined using the R HMR package to plot best-fit lines to
the data for headspace concentration against time for individual flux chambers (Pedersen et
al., 2010;R Core Team, 2012). Gas mixing ratios (ppm) were converted to areal flux by using
the Ideal Gas Law to solve for the quantity of gas in the headspace (on a mole or mass basis),
normalized by the surface area of each static flux chamber (Livingston and Hutchinson,

192    1995).


**4.3 Environmental variables**
To investigate the effects of environmental variables on trace gas dynamics, we determined
soil moisture, soil oxygen content in the 0-10 cm depth, soil temperature, and air
temperature at the time of flux sampling. Volumetric soil moisture content was determined
using portable soil moisture probes (ML2x ThetaProbe, Delta-T Device Ltd., Cambridge, UK)
inserted into the substrate immediately adjacent to each flux chamber (<5 cm from each
chamber base; depth of 0-10 cm). Soil moisture content is reported here as water-filled pore
space (WFPS), and is calculated using the measurements of volumetric water content and
bulk density (Breuer et al., 2000). Soil $O_2$ concentration was determined using the approach
described by Teh et al. (2014). Soil temperature (0-10 cm depth), chamber temperature and
air temperature was determined using type K thermocouples (Omega Engineering Ltd.,
Manchester, UK). Data on aboveground litter-fall, meteorological variables (i.e.
photosynthetically active radiation, air temperature, relative humidity, rainfall, wind speed,
wind direction), continuous plot-level soil moisture (10 and 30 cm depths) and soil





temperature (0, 10, 20 and 30 cm depths) measurements were also collected, but are not
reported in this publication.

Resin-extractable inorganic N flux (i.e. ammonium, $NH_4^+$; nitrate, $NO_3^-$) were quantified in all
plots using a resin bag approach (Templer et al., 2005;Subler et al., 1995). From August 2011
onwards, ion exchange resin bags (n = 15 resin bags per elevation) were deployed at the
bottom of the plant rooting zone (i.e. 0-10 cm depth in premontane forest, lower montane
forest and montane grasslands; 0-15 cm in upper montane forest), following established
protocols (Templer et al., 2005;Subler et al., 1995). Samples were collected at monthly
intervals (where possible) for determination of monthly, time-averaged $NH_4^+$ and $NO_3^-$ flux
(Subler et al., 1995). For some plots, this sampling frequency was periodically disrupted due
to natural hazards (i.e. landslides, river flooding) preventing safe access to the study sites.
Resin bags were shipped to the University of Aberdeen after collection from the field,
inorganic N was extracted using 2 M KCl and concentrations determined colourimetrically
using a Burkard SFA2 continuous-flow analyser (Burkard Scientific Ltd., Uxbridge, UK)
(Templer et al., 2005;Subler et al., 1995).

**4.4 Water-filled pore space manipulation study**
We investigated the effects of WFPS on $N_2O$ flux derived from nitrate reduction or
nitrification rates using a [15]N tracer experiment. Soil cores for all habitats were collected
from the 0-10 cm depth, distributed into glass jars and adjusted to 10% below the target
WFPS values of 30%, 50%, 70% and 90% (n = 5 for each [15]N addition and 3 controls for each
WFPS for a total of n = 212; see Table 2). Additional de-ionized water was added
gravimetrically to raise WFPS to target levels. The exception to this was for the upper
montane forest, where samples were collected from the 0-10 cm depth of the mineral soil,
but not from the organic layer. Two different types of [15]N-tracers were applied to the soils in
order to determine the proportion of $N_2O$ derived from nitrate reduction and nitrification
(Bateman and Baggs, 2005). $^{14}N\text{-}NH_4\,^{15}N\text{-}NO_3$ was used to quantify the amount of $N_2O$
produced by nitrate reduction, while $^{15}N\text{-}NH_4\,^{15}N\text{-}NO_3$ was used to quantify the amount of
$N_2O$ produced from both nitrate reduction and nitrification. The difference between the two
was used to calculate the amount of $N_2O$ derived from nitrification alone. After application
of the tracers, the jars were sealed, and gas samples taken at 0, 6, 12, 24, 36 and 48 hours to





determine rates of gas flux. Nitrous oxide yield was calculated as the ratio of $^{15}N$-$N_2O$ flux :
$^{15}N$-$N_2O$ flux + $^{15}N$-$N_2$ flux. Soils were sampled at the end of the experiment for $NO_3^-$
concentration, $NH_4^+$ concentraion, and total C and N content.

Soil gas concentrations ($N_2O$, $CO_2$ and $CH_4$) were measured on a GC as described in section
4.2, while $^{15}N$-$N_2$ and $^{15}N$-$N_2O$ were measured on a SerCon 20:20 isotope ratio mass
spectrometer equipped with an ANCA TGII pre-concentration module (SerCon Ltd., UK). The
coefficient of variation (CV; an index of instrumental precision) for repeated analysis of gas
concentration and isotope standards was <5 %. $^{15}N$-$N_2O$ and $^{15}N$-$N_2$ fluxes were calculated
from the $^{15}N$ atom percent excess of the samples compared to the controls using the HMR
package (Pedersen et al., 2010). Nitrous oxide yield was calculated as the ratio of $^{15}N$-$N_2O$
flux : $^{15}N$-$N_2O$ flux + $^{15}N$-$N_2$ flux.

**4.5 Litter-fall manipulation experiments**
We conducted a field-based litter-fall manipulation experiment to test for the effects of
variations in labile organic matter availability on trace gas flux. This study took place over a
14-month period (April 2012 to June 2013), and consisted of 4 experimental treatments
(control, +50 % litter addition, +100 % litter addition, litter removal) implemented across 3
habitats (premontane forest, lower montane forest, upper montane forest), with 6 replicate
plots per treatment per habitat (each treatment plot was 0.5 x 0.5 m in size; n = 24
observations per habitat; n = 72 observations per sampling increment). Leaf litter addition
rates for the +50 % and +100 % litter addition treatments were determined based on prior
research from this study site, and fell within the natural range of variability observed across
this elevational gradient (Girardin et al., 2010).

Litter-fall for the litter addition treatments was collected monthly in litter baskets (n = 3
litter baskets per treatment plot for a total of n = 18  per habitat). These data were also used
to determine the background rates of leaf litter-fall among habitats. For the control, litter
inputs simply reflected natural background litter-fall rates. For the +50 % and +100 % litter
addition treatments, background litter inputs were supplemented with additional litter
taken from the litter baskets. Briefly, wet litter was weighed in the field using portable scale,
gently mixed (homogenized), and then re-distributed to the +50 % and +100 % litter addition



plots in amounts proportional to the average amount of wet litter that fell into the litter
baskets over the course of the month. As a consequence, the amount of litter added in the
two litter addition treatments was not fixed but varied according to the natural background
rate of litter-fall. For the litter removal treatment, leaf litter was removed from the forest
floor at the start of the experiment, and 3mm nylon mesh was placed over the surface of the
treatment plot to prevent further litter ingress to the soil surface. Any debris accumulating
on the mesh was removed at monthly intervals.

Trace gas flux and environmental variables were determined at 7 time points over the
course of the 14-month experiment using the methods described in section 4.2. In addition,
soil moisture (WFPS from the 0-10 cm depth), soil temperature (0-10 cm depth), air
temperature, soil gas concentrations ($O_2$, $CH_4$, $N_2O$, $CO_2$) from the 0-10 cm and 20-30 cm
depths, litter C, and litter N were determined concomitantly. Litter C and N content was
determined on a Carlo-Erba NA 2500 elemental analyser (CE Instruments Ltd, Wigan, UK) at
the University of Aberdeen.

**4.6 Nitrate addition experiment**
To quantify the effect of $NO_3^-$ availability on $N_2O$ flux, we conducted a $^{15}N$-$NO_3^-$ addition
experiment. Background concentrations of $NO_3^-$ were determined prior to the start of
experiment using soil subsamples, after which the soils from each habitat were divided into
three treatment groups, and supplemented with surplus $NO_3^-$ which raised these
background levels by +50 %, +100 %, and +150 % (Table 2). The $NO_3^-$ added to the soil in
each of the treatments was enriched with $^{15}N$ in order to trace the conversion of nitrate to
gaseous N products ($^{15}N$-$N_2O$, $^{15}N$-$N_2$) (Baggs, 2003;Bateman and Baggs, 2005).

Soil cores were sampled from 0-10 cm for each habitat (n = 6 soil cores per habitat), with the
exception for upper montane forest, where two separate sets of cores were collected, one
from the organic layer (O horizon; n = 6) and the other from the mineral layer (A horizon; n =
6). Soil samples were then shipped to the University of Aberdeen. Five of these soil cores
were split into four equal parts (3 treatment cores and one control core) and distributed into
1 L screw top jars (Kilner, UK). A small soil subsample from each core was used to determine
WFPS, background $NO_3^-$ content (extracted in 100ml 1M KCl for a 10g soil sample prior to the



start of the experiment), as well as total C and N content. If necessary, the cores were
gravimetrically amended with water until the cores reached 80% WFPS. Soil cores were kept
under constant conditions for 3 days before the start of the experiment to minimise the
effects of changing water content on soil processes.

At the start of the experiment, dissolved $^{15}$N-labelled $KNO_3$ (30 atom %) was added
according to the measured $NO_3^-$ concentrations of each core to reach the required $NO_3^-$
concentration for each treatment (Table 2). Initial $NO_3^-$ concentration (prior to $^{15}$N addition)
averaged (± standard error) 157 ± 12 µg N g soil$^{-1}$ for pre-montane forest, 140 ± 12 µg N g
soil$^{-1}$ for lower montane forest, 19 ± 7 µg N g soil$^{-1}$ for upper montane forest organic layer
soil, 18 ± 5 µg N g soil$^{-1}$ for upper montane forest mineral layer soil, and 6 ± 2 µg N g soil$^{-1}$ for
montane grassland soil (Table 2). The jars were then sealed with lids fitted with a two-way
stopcock to allow for gas sampling. Gas samples were taken with gas tight syringes, and
stored in pre-evacuated containers for determination of $^{15}$N-$N_2$, $^{15}$N-$N_2O$, $N_2O$, $CO_2$ and $CH_4$
content. Isotope samples (150 ml) were stored in 100 mL serum bottles and gas
concentration samples (20 ml) were stored in 12 ml Exetainers® (Labco Ltd., Lampeter, UK).
After gas sampling, the stopcock was opened to allow the sampled air from the jar to be
replaced by lab air, and lab air was sampled to allow for correction of the gas concentrations
in the jars due to dilution. Samples were taken at 0, 6, 12, 24, 36, and 48 hours, after which
the jars were opened and soil was sampled for determination of $NO_3^-$, $NH_4^+$ and total C and
N. Gas flux, isotopic and elemental concentrations were determined according to the
methods described previously.

**4.7 Statistics**
Statistical analyses were performed using JMP IN Version 8 (SAS Institute, Inc., Cary, North
Carolina, USA) or R (R Core Team, 2012). Residuals were checked for heteroscedasticity and
homogeneity of variances. Where necessary, the data were transformed using a Box-Cox
procedure to meet the assumptions of analysis of variance. Analysis of variance (ANOVA) or
Generalized Linear Models were used to evaluate the effect of categorical variables (i.e. site,
season, topography) on trace gas flux and environmental variables. Analysis of covariance
(ANCOVA) was performed on Box-Cox transformed data to investigate the combined effects
of categorical variables and environmental factors (e.g. water-filled pore space, soil oxygen





content, air temperature, soil temperature, etc.) on trace gas flux. Non-parametric tests
were employed where Box-Cox transformation was unable to normalize the data,
homogenize the variances, or where the residuals still showed strong trends even after Box-
Cox transformation. Means comparisons were performed using Fisher's Least Significant
Difference test (Fisher's LSD). Statistical significance was determined at the $P < 0.05$ level,
unless otherwise noted. Values are reported as means and standard errors (± 1 SE).
Statistical analyses for the field data were conducted on plot-averaged data to avoid pseudo-
replication.


**5. Results**
**5.1 Variations in $N_2O$ flux among habitats and between seasons**
The overall mean $N_2O$ flux for the entire dataset was $0.27 ± 0.07$ mg $N-N_2O$ $m^{-2}$ $d^{-1}$, with a
range from -8.40 to 75.0 mg $N-N_2O$ $m^{-2}$ $d^{-1}$. We investigated the effect of habitat, season,
and topography on $N_2O$ flux by using a three-way ANOVA on plot-averaged data ($F_{10,307}$ =
3.28, $P < 0.0005$). We found that there was a significant effect of habitat ($P < 0.003$) and an
effect of season at the borderline of statistical significance ($P < 0.07$). However, we found no
effect of habitat by season or topography on $N_2O$ flux. Habitat accounted for 4.3 % of the
variance in the dataset, while season accounted for only 1.0 % of the variance.

Among habitats, the overall trend was towards the highest flux from premontane forest
($0.75 ± 0.18$ mg $N-N_2O$ $m^{-2}$ $d^{-1}$), followed by lower montane forest ($0.46 ± 0.24$ mg $N-N_2O$ $m^{-2}$
$d^{-1}$), montane grasslands ($0.07 ± 0.08$ mg $N-N_2O$ $m^{-2}$ $d^{-1}$), and upper montane forest ($0.04 ±$
$0.07$ mg $N-N_2O$ $m^{-2}$ $d^{-1}$) (Figure 2a). Multiple comparisons tests indicated that only
premontane forests showed statistically higher flux than the others (Fisher's LSD, $P < 0.05$);
while there were numerical differences in mean flux among the other habitats, large
variances meant that they had overlapping ranges of flux (Figure 2a).

The borderline significant effect of season ($P < 0.07$) reflected an overall trend of higher dry
season ($0.51 ± 0.18$ mg $N-N_2O$ $m^{-2}$ $d^{-1}$) compared to wet season ($0.15 ± 0.07$ mg $N-N_2O$ $m^{-2}$ $d^{-}$
$^1$) flux (Table 3). However, part of why the effect of season was weak was because only lower
montane forest showed significant variability between seasons (Fisher's LSD, $P < 0.05$), while





the other three habitats did not show significant seasonal differences in flux (Fisher's LSD, $P$
< 0.05).

Even though the effect of topography alone was not statistically significant within the
context of the three-way ANOVA, $N_2O$ flux from flat sites were significantly higher (0.62 ±
0.28 mg N-$N_2O$ m$^{-2}$ d$^{-1}$) than from the basin site (-0.18 ± 0.16 mg N-$N_2O$ m$^{-2}$ d$^{-1}$) (Fisher's LSD,
$P$ < 0.05). However, there was no significant difference between flat sites with slope and
ridge sites (0.24 ± 0.09 mg N-$N_2O$ m$^{-2}$ d$^{-1}$ and 0.20 ± 0.08 mg N-$N_2O$ m$^{-2}$ d$^{-1}$, respectively)
(Fisher's LSD, $P$ > 0.05).

For each habitat, we also compared individual wet and dry seasons against each other using
multiple comparisons tests (e.g. dry season 2012 vs wet season 2012; dry season 2012 vs dry
season 2013, etc.) to determine if there was significant year-on-year variation in $N_2O$ flux
among multiple seasons. Consistent with our three-way ANOVA results, we found that only
lower montane forest showed significant variation among multiple dry and wet seasons,
whereas the other habitats showed no significant trends. For lower montane forest, we
observed significantly higher dry season flux in 2011 compared to wet and dry seasons in all
other years ($P$ < 0.05; Figure 3b).

**5.2 Variations in environmental conditions among habitats and between seasons**
We investigated the effect of habitat, season, and topography on environmental variables by
using a three-way ANOVA on plot-averaged data. The environmental variables examined
here were water-filled pore space (WFPS) in the 0-10 cm depth, soil temperature, air
temperature, gas-phase soil oxygen content in the 0-10 cm depth, and resin-extractable
inorganic N flux ($NH_4^+$, $NO_3^-$).

Water-filled pore space varied significantly as a function of habitat, season, habitat by
season, and topography ($F_{10,304}$ = 637.96, $P$ < 0.0001; Table 3, Figure 2b, Figure 3). Habitat
accounted for the largest proportion of variance in the model (78.1 % of the total variance),
followed by season (0.6 %), habitat by season interaction (0.6 %), and topography (0.4 %).
Each habitat differed significantly from the others (Fisher's LSD, $P$ <0.05), with the highest
WFPS observed in montane grassland (88.4 ± 0.3 %), followed by premontane forest (51.6 ±

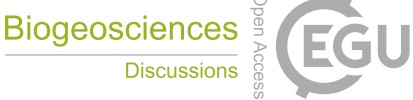

1.3 %), lower montane forest (39.0 ± 0.9 %), and upper montane forest (35.0 ± 1.5 %) (Figure
2b). WFPS varied significantly between seasons (t-Test, $P < 0.05$), with a mean dry season
value of 52.1 ± 2.4 % compared to a mean wet season value of 59.5 ± 1.6 % (Table 3). The
significant habitat by season interaction is due to the fact that some habitats showed
seasonal trends in WFPS whereas others did not. Whereas lower montane and upper
montane forests all showed a significant reduction in WFPS during the dry season,
premontane forest and montane grasslands showed no seasonal differences in WFPS (Table
3, Figure 3). For topography, the main effect was that the basin landform had significantly
higher WFPS than the other landforms. The basin landform showed a mean WFPS of 89.3 ±
0.1 % whereas WFPS in other landforms ranged from 51.7 ± 2.2 to 57.7 ± 2.7 %.

Soil oxygen in the 0-10 cm depth varied as a function of habitat, habitat by season, and
topography ($F_{10,242} = 27.70$, $P < 0.0001$; Table 3). The effect of season was significant at the $P$
< 0.06 level. Habitat accounted for the largest proportion of variance in the model (66.9 % of
the total variance), followed by topography (8.4 %), habitat by season (3.5 %), and season
alone (0.7 %). For habitat, multiple comparisons tests indicated that montane grasslands
showed significantly lower soil $O_2$ content than the other habitats (13.5 ± 0.6 %), whereas
the other habitats showed statistically similar soil $O_2$ values to each other (18.6 ± 0.2 to 19.5
± 0.1 %; Fisher's LSD, $P < 0.05$). For topography, multiple comparisons tests indicated that
the basin landform showed statistically lower soil $O_2$ content than the other landforms (7.4 ±
2.3 %), whereas the other topographic features showed statistically similar values, ranging
from 16.9 ± 0.6 to 18.2 ± 0.2 % (Fisher's LSD, $P < 0.05$). The significant habitat by season
interaction was due to the fact that only montane grassland showed a significant difference
in $O_2$ content between wet and dry season, whereas other habitats showed similar soil $O_2$
values (Table 3). For season alone, wet season soil $O_2$ content (16.8 ± 0.4 %) was slightly
lower than dry season values (17.8 ± 0.3 %) (t-Test, $P < 0.03$); however, given the significant
habitat by season interaction described previously, this weak seasonal trend in the pooled
dataset was likely driven by the seasonal pattern in montane grassland.

For soil temperature, the effects of habitat, season, habitat by season, and topography were
all significant ($F_{10,292} = 790.7$, $P < 0.0001$). Habitat accounted for the largest proportion of
variance in the model (85.5 % of the total variance), followed by season (1.4%), habitat by





season interaction (0.5 %), and topography (0.3 %). Each habitat differed significantly from
the others (Fisher's LSD, $P$ <0.05), with the highest soil temperature observed for
premontane forest (20.5 ± 0.1 °C), followed by lower montane forest (17.8 ± 0.1 °C), upper
montane forest (11.5 ± 0.1 °C), and montane grasslands (10.6 ± 0.2 °C). Soil temperature
varied significantly between season (t-Test, $P < 0.05$), with a mean dry season value of 13.9 ±
0.4 °C compared to a mean wet season value of 15.1 ± 0.3 °C. The significant habitat by
season interaction is due to the fact that some habitats showed more pronounced seasonal
trends in soil temperature than others, although the overall pattern of cooler dry season
compared to wet season soil temperatures holds across all habitats (Table 3). For
topography, the flat landforms showed significantly higher soil temperatures than the others
(16.0 ± 0.5 °C), the basin landform showed significantly lower values (10.8 ± 0.4 °C), whereas
ridge and slope landforms showed similar values to each other (14.3 ± 0.4 °C and 14.7 ± 0.4
°C, respectively) (Fisher's LSD, $P < 0.05$).

For air temperature, only the effect of habitat was significant ($F_{10,292}$ = 103.2, $P < 0.0001$;
Table 3). A multiple comparisons test indicated that each habitat showed significantly
different temperatures compared to the others (Fisher's LSD, $P < 0.05$). Premontane forest
showed the highest air temperatures (21.0 ± 0.3 °C), followed by lower montane forest (18.7
± 0.2 °C), upper montane forest (12.7 ± 0.2 °C), and montane grassland (11.7 ± 0.3 °C). Other
variables did not significantly affect air temperature.

For resin-extractable $NH_4^+$ flux, the three-way ANOVA model was not statistically significant
($F_{10,164}$ = 1.3, $P > 0.2$; Table 3). However, even though the three-way ANOVA as a whole was
not statistically significant, the overall trend was towards significantly lower $NH_4^+$ flux in the
dry season (9.6 ± 0.7 µg N-NH$_4$ g resin$^{-1}$ d$^{-1}$) compared to the wet season (22.3 ± 3.6 µg N-NH$_4$
g resin$^{-1}$ d$^{-1}$).

Resin-extractable $NO_3^-$ flux showed different patterns from $NH_4^+$ flux, with significant effects
of habitat, topography, and habitat by season but not of season alone ($F_{10,164}$ = 39.0, $P <$
0.0001; Figure 2c, Table 3). Habitat accounted for the largest proportion of the variance
(61.5 %), followed topography (4.7 %), and habitat by season (1.9 %). Premontane forest
showed the highest $NO_3^-$ flux (22.6 ± 2.0 µg N-NO$_3$ g resin$^{-1}$ d$^{-1}$), followed by lower montane





forest (10.0 ± 1.2 µg N-NO$_3$ g resin$^{-1}$ d$^{-1}$) (Fisher's LSD, $P$ < 0.05; Figure 2c). Upper montane
forest (1.1 ± 0.2 µg N-NO$_3$ g resin$^{-1}$ d$^{-1}$) and montane grassland (1.7 ± 0.3 µg N-NO$_3$ g resin$^{-1}$ d$^{-}$
$^1$) showed significantly lower NO$_3^-$ flux than the other two habitats (Fisher's LSD, $P$ < 0.05;
Figure 2c). However, NO$_3^-$ flux in upper montane forest and montane grassland did not differ
significantly from each other (Fisher's LSD, $P$ > 0.05; Figure 2c). For the effect of topography,
multiple comparisons tests indicated that flat landforms (12.1 ± 1.8 µg N-NO$_3$ g resin$^{-1}$ d$^{-1}$)
and slope landforms (10.2 ± 1.6 µg N-NO$_3$ g resin$^{-1}$ d$^{-1}$) differed significantly from ridge
landforms (6.6 ± 1.4 µg N-NO$_3$ g resin$^{-1}$ d$^{-1}$) (Fisher's LSD, $P$ < 0.05). The basin landform (3.8 ±
1.3 µg N-NO$_3$ g resin$^{-1}$ d$^{-1}$), despite the lower mean values, showed an overlapping range
with the other landforms (Fisher's LSD, $P$ > 0.05). The habitat by season interaction is due to
the fact that upper montane forest shows a significant seasonal fluctuation in resin-
extractable NO$_3^-$ (Fisher's LSD, $P$ < 0.05), whereas the other habitats show no significant
seasonal trend (Fisher's LSD, $P$ > 0.05).

**5.3 Effects of environmental variables on N$_2$O flux**
For the whole dataset, the relationship between N$_2$O flux and environmental variables was
examined using ANCOVA on Box-Cox transformed data with habitat, season, topography,
and environmental variables as covariates. Environmental variables included WFPS, oxygen,
air temperature, soil temperature, and resin-extractable inorganic N flux (NH$_4^+$ and NO$_3^-$).
The ANCOVA model as a whole was not statistically significant ($P$ > 0.4). However, we found
that individual factors were weakly but significantly correlated with N$_2$O flux for the pooled
dataset. These included soil temperature (r$^2$= 0.04, $P$ <0.0004), air temperature (r$^2$= 0.04, $P$
<0.0008), and resin-extractable NO$_3^-$ flux (r$^2$= 0.03, $P$ <0.03). Water-filled pore space also
showed a very weak negative correlation with N$_2$O flux at the borderline of statistical
significance (r$^2$= 0.01, $P$ <0.06).

For individual habitats, we explored how variations in environmental conditions influenced
N$_2$O flux using multiple regression, with WFPS, oxygen, soil temperature, air temperature,
resin-extractable NH$_4^+$ flux, and resin-extractable NO$_3^-$ flux as explanatory variables. Only the
multiple regression analysis for lower montane forest showed a borderline significant result,
though only at the $P$ < 0.07 level ($r^2$ = 0.36). The multiple regression models for all the other
habitats were not statistically significant ($P$ > 0.4). Lower montane forest was the only



habitat that showed a significant effect of season on $N_2O$ flux (section 5.1), and our multiple
regression model corroborated this result by showing that seasonal fluctuations in air
temperature, soil temperature, WFPS (Figure 3b), and $NH_4^+$ all correlated with $N_2O$ flux ($P <$
0.05). Air temperature explained the largest proportion of variance in the data (26.2 %;
negative trend), followed by soil temperature (15.5 %; positive trend), WFPS (13.7 %;
negative trend), and resin-extractable $NH_4^+$ flux (11.6 %; negative trend).

**5.4 Water-filled pore space manipulation**
$^{15}N$-$N_2O$ and $^{15}N$-$N_2$ fluxes showed a biphasic response (Limmer and Steele, 1982), with
significantly different flux rates in the first 24 hours of incubation compared to the later
period of incubation (i.e. >24 hours onwards). Flux of $^{15}N$-$N_2O$, and $^{15}N$-$N_2$ were therefore
divided into early (≤24 hours) and late (>24 hours) phase flux.

**5.4.1 Role of nitrate reduction in $N_2O$ production**
For both the $^{15}N$-$N_2O$ and $^{15}N^-N_2$ flux data, we conducted an initial analysis using a full
factorial ANOVA on Box-Cox transformed data with habitat, moisture level, form of $^{15}N$-label
added (i.e. $^{15}NH_4^{15}NO_3$ or $^{14}NH_4^{15}NO_3$), incubation phase, and all their interaction terms as
independent variables. Importantly, we found that the form of $^{15}N$-label added (i.e. $^{15}N$-
$NH_4^{15}N$-$NO_3$ or $^{14}N$-$NH_4^{15}N$-$NO_3$) did not significantly influence $^{15}N$-$N_2O$ or $^{15}N$-$N_2$ flux,
because production of either gas from $^{15}N$-$NH_4^{15}N$-$NO_3$ addition was modest to negligible
(Supplementary Online Materials Figure S1). This indicates that that nitrate reduction was
the dominant source of $N_2O$ among these habitats. Thus, in order to simplify our statistical
analyses, all subsequent analyses were performed using only habitat, moisture level,
incubation phase, and their interaction terms as independent variables. For these tests,
which are described below, the "total" flux of $^{15}N$-$N_2O$ or $^{15}N$-$N_2$ represents gas produced by
both nitrification and nitrate reduction together.

**5.4.2 $^{15}N$-$N_2O$ flux**
For the total $^{15}N$-$N_2O$ flux data, we used a full factorial ANOVA on Box-Cox transformed data
with habitat, moisture level, incubation phase, and all their interactions as independent
variables. We found that moisture level, habitat by incubation phase, and habitat by
moisture by incubation phase significantly affected flux, while all other factors were not




statistically significant (ANOVA, $F_{31,\,321}$ = 3.05, $P$ < 0.0001; Figure 4). For the moisture level
effect, the highest flux was observed for the 90 % WFPS (42 ± 9 ng $N_2O$-$^{15}N$ $g^{-1}$ $d^{-1}$) and 50 %
WFPS (29 ± 10 ng $N_2O$-$^{15}N$ $g^{-1}$ $d^{-1}$) treatments, and the lowest flux for the 30 % (3 ± 1 ng $N_2O$-
$^{15}N$ $g^{-1}$ $d^{-1}$) and 70 % (7 ± 2 ng $N_2O$-$^{15}N$ $g^{-1}$ $d^{-1}$) treatments (Fisher's LSD, $P$ < 0.05; Figure 4).

The habitat by incubation phase interaction indicated that some habitats showed different
flux from each other during different phases of the incubation (Figure 4). For example,
premontane and lower montane forest showed no significant difference in flux during
different incubation phases (t-Test, $P$ > 0.05 for each habitat), whereas upper montane
forest mineral layer soils showed a significant increase from early to late incubation phases
(5 ± 2 ng $N_2O$-$^{15}N$ $g^{-1}$ $d^{-1}$ versus 42 ± 13 ng $N_2O$-$^{15}N$ $g^{-1}$ $d^-$1; t-Test, $P$ < 0.003). In contrast to
the other habitats, montane grasslands showed a significant decrease in flux from early to
late incubation phases (60 ± 23 ng $N_2O$-$^{15}N$ $g^{-1}$ $d^{-1}$ versus 6 ± 9 ng $N_2O$-$^{15}N$ $g^{-1}$ $d^{-1}$, respectively;
t-Test, $P$ < 0.02).

The habitat by moisture by incubation phase effect indicated that different habitats showed
varying responses to moisture depending on the incubation phase (Figure 4). For example,
for the premontane and lower montane forest, which showed no effect of incubation phase,
flux followed the moisture trend described for the data set as a whole (i.e. highest flux for
the 90 % WFPS treatment, lowest flux for the 30 % WFPS treatment, intermediate flux for
the 50 & 70 % WFPS treatments). In contrast, for upper montane forest mineral layer soils,
the effects of moisture varied with incubation phase. During the early phase, flux was
highest in the 50 % WFPS treatment (20 ± 8 ng $N_2O$-$^{15}N$ $g^{-1}$ $d^{-1}$), while all other treatments
showed lower flux (pooled average of 0.5 ± 0.4 ng $N_2O$-$^{15}N$ $g^{-1}$ $d^{-1}$). In the late phase, flux was
highest for the 90 % WFPS treatment (145 ± 40 ng $N_2O$-$^{15}N$ $g^{-1}$ $d^{-1}$) while the other
treatments were lower and not statistically different from each other (pooled average: 13 ±
5 ng $N_2O$-$^{15}N$ $g^{-1}$ $d^{-1}$)

### 556 5.4.3 $^{15}N$-$N_2$ flux

For the total $^{15}N$-$N_2$ flux data, we used a full factorial ANOVA on Box-Cox transformed data
with habitat, moisture level, incubation phase, and all their interactions as independent
variables. We found that all of the main factors and their interaction terms were statistically





significant (ANOVA, $F_{31, 317}$ = 14.20, $P < 0.0001$). For the habitat effect, lower montane forest
had the highest flux (694 ± 83 ng $N_2$-$^{15}N$ $g^{-1}$ $d^{-1}$), while premontane forest and upper
montane forest mineral layer collectively had intermediate flux soil (326 ± 53 and 171 ± 20
ng $N_2$-$^{15}N$ $g^{-1}$ $d^{-1}$, respectively) (Fisher's LSD, $P < 0.05$; Figure 4). Montane grassland soil had
the lowest flux (123 ± 23 ng $N_2O$-$^{15}N$ $g^{-1}$ $d^{-1}$) (Fisher's LSD, $P < 0.05$; Figure 4). For the
moisture effect, only the 90 % treatment had significantly higher flux than the other
treatments (90 % WFPS treatment: 437 ± 77 ng $N_2$-$^{15}N$ $g^{-1}$ $d^{-1}$; pooled average for all other
treatments: 294 ± 28 ng $N_2$-$^{15}N$ $g^{-1}$ $d^{-1}$) (Fisher's LSD, $P < 0.05$). The effect of incubation phase
was only significant at the $P < 0.1$ level, with greater release of $^{15}N$-$N_2$ during the late
compared to the early phase of the incubation (373 ± 44 ng $N_2$-$^{15}N$ $g^{-1}$ $d^{-1}$ versus 288 ± 37 ng
$N_2$-$^{15}N$ $g^{-1}$ $d^{-1}$) (t-Test, $P < 0.07$).

The habitat by moisture level interaction indicates that flux from different habitats showed
varying moisture responses (Figure 4). For example, flux from premontane forest and upper
montane forest mineral layer soil showed no responses to moisture. In contrast, for lower
montane forest, flux was greatest for the 90 % WFPS treatment (1,365 ± 201 ng $N_2$-$^{15}N$ $g^{-1}$ $d^{-1}$
), lowest for the 70 % WFPS treatment (257 ± 128 ng $N_2$-$^{15}N$ $g^{-1}$ $d^{-1}$), and at intermediate
levels for the 30 and 50 % WFPS treatments (664 ± 131 and 492 ± 79 ng $N_2$-$^{15}N$ $g^{-1}$ $d^{-1}$,
respectively) (Fisher's LSD, $P < 0.05$). The pattern for montane grassland was different again;
here, only the 90 % WFPS treatment showed significantly greater flux (171 ± 32 ng $N_2$-$^{15}N$ $g^{-1}$
$d^{-1}$) compared to the other treatments (pooled average: 105 ± 29 ng $N_2$-$^{15}N$ $g^{-1}$ $d^{-1}$) (Fisher's
LSD, $P < 0.05$).

The habitat by incubation phase interaction indicates that flux for different habitats showed
different patterns during early and late incubation phases (Figure 4). For example,
premontane forest showed a significant increase for early (169 ± 42 ng $N_2$-$^{15}N$ $g^{-1}$ $d^{-1}$) to late
(483 ± 91 ng $N_2$-$^{15}N$ $g^{-1}$ $d^{-1}$) incubation phases (t-Test, $P < 0.01$. In contrast, lower montane
forest, upper montane forest mineral layer soil, and montane grassland all showed no
significant change in flux between incubation phases (t-Test, $P > 0.05$ for all habitats).

Finally, the habitat by moisture level by incubation phase interaction indicates that moisture
responses among habitats were influenced by incubation phase (Figure 4). For example, for





the premontane forest, where an incubation phase effect was found, the response to
moisture varied depending on incubation phase. During the early phase of the incubation,
flux was lowest from the 70 % WFPS treatment ($0 \pm 0$ ng $N_2$-$^{15}$N $g^{-1}$ $d^{-1}$), while all other
moisture treatments showed similar levels of flux (pooled average: $224 \pm 52$ ng $N_2$-$^{15}$N $g^{-1}$ $d^{-1}$
). For the late phase, the highest flux was observed for the 70 % WFPS treatment ($1,267 \pm$
$175$ ng $N_2$-$^{15}$N $g^{-1}$ $d^{-1}$), followed by the 50 % WFPS treatment ($540 \pm 99$ ng $N_2$-$^{15}$N $g^{-1}$ $d^{-1}$), the
90 % treatment ($157 \pm 43$ ng $N_2$-$^{15}$N $g^{-1}$ $d^{-1}$), and the 30 % WFPS treatment ($0 \pm 0$ ng $N_2$-$^{15}$N $g^{-1}$
$d^{-1}$) (Fisher's LSD, $P < 0.05$). In contrast, for all other habitats, where there was no significant
incubation phase effect (i.e. lower montane forest, upper montane forest mineral layer soil,
montane grassland), the response to moisture followed the overall pattern described
previously.

**5.4.4 $N_2O$ Yield**
For the $N_2O$ yield, we used a full factorial ANOVA on Box-Cox transformed data with habitat,
moisture level, incubation phase, and all their interactions as independent variables. We
found that habitat, moisture level, habitat by moisture level, habitat by phase, and habitat
by moisture level by phase significantly influenced $N_2O$ yield (ANOVA, $F_{31, 313} = 9.85$, $P <$
$0.0001$). For the habitat effect, $N_2O$ yield was highest for the montane grassland ($0.61 \pm$
$0.06$), lowest for lower montane forest ($0.19 \pm 0.04$), while premontane forest and upper
montane forest mineral layer soil showed similar intermediate values ($0.40 \pm 0.05$ and $0.42 \pm$
$0.05$, respectively) (Fisher's LSD, $P < 0.05$). For the moisture level effect, $N_2O$ yield was
highest for the 70 % WFPS treatment ($0.51 \pm 0.06$), while the 30, 50 and 90 % WFPS
treatments showed statistically similar values ($0.35 \pm 0.05$, $0.39 \pm 0.05$, and $0.36 \pm 0.05$,
respectively) (Fisher's LSD, $P < 0.05$).

The interaction terms indicated that different habitats showed varying $N_2O$ yield in response
to moisture level and incubation phase. For the habitat by moisture level interaction, some
habitats showed no effect of moisture level on $N_2O$ yield (i.e. premontane forest, montane
grassland), whereas others showed changes in $N_2O$ yield with moisture level. For example,
for the lower montane forest, $N_2O$ yield was greatest for the 70 % WFPS treatment ($0.51 \pm$
$0.11$), whereas the 30, 50 and 90 WFPS % treatments were statistically undifferentiated from
each other (pooled average: $0.09 \pm 0.03$) (Fisher's LSD, $P < 0.05$). Upper montane forest



mineral layer soil showed the highest $N_2O$ yield for the 90 % treatment (0.72 ± 0.08), lowest
yield for the 30 % WFPS treatment (0.20 ± 0.09), and intermediate $N_2O$ yields for the 50 and
70 % WFPS treatments (0.29 ± 0.09 and 0.50 ± 0.11, respectively) (Fisher's LSD, $P < 0.05$). For
the habitat by phase interaction, some habitats showed no effect of incubation phase on
$N_2O$ yield (i.e. premontane and lower montane forest), whereas some showed an increase in
$N_2O$ yield from early to late phase (i.e. upper montane forest mineral layer soil), while still
others showed a decrease in $N_2O$ yield from early to late phase (i.e. montane grassland). For
the upper montane forest mineral layer soil, $N_2O$ yield shifted from 0.33 ± 0.07 to 0.51 ± 0.07
(t-Test, $P < 0.04$), while for montane grassland $N_2O$ yield changed from 0.70 ± 0.07 to 0.52 ±
0.09 (t-Test, $P < 0.05$).

The habitat by moisture level by incubation phase interaction reflects the fact that the
moisture response of different habitats was contingent upon incubation phase. For instance,
for upper montane forest mineral layer soil, $N_2O$ yield during the early phase was greatest
for the 90 % WFPS treatment (1; i.e. no [15]$N$-$N_2$ flux observed), while the 50 % WFPS
treatment showed intermediate $N_2O$ yield (0.33 ± 12), and the 30 and 70 % WFPS treatments
collectively showed the lowest $N_2O$ yields (approximately 0 for both; i.e. no [15]$N$-$N_2O$ flux
observed) (Fisher's LSD, $P < 0.05$). In contrast, during the late phase, the 70 % WFPS
treatment showed the highest $N_2O$ yield (1; i.e. no [15]$N$-$N_2$ flux observed), while the other
treatments showed lower $N_2O$ yields that were not significantly different from each other
(pooled average: 0.33 ± 0.07) (Fisher's LSD, $P < 0.05$). In contrast, for montane grassland, no
effect of moisture was observed during the early phase of the incubation. However, during
the late phase, the 50 % WFPS treatment showed the highest $N_2O$ yield (0.89 ± 0.11), while
the other treatments showed lower $N_2O$ yields that were not significantly different from
each other (pooled average: 0.39 ± 0.10) (Fisher's LSD, $P < 0.05$). For all other habitats with
no habitat by phase interaction (i.e. premontane and lower montane forest), the moisture
effect follows the general trends described above.

**5.5 Litter manipulation experiment**
In order to investigate the relationship between leaf litter input rates and $N_2O$ flux, we used
a Generalized Linear Model (GLM) and an ANCOVA that included habitat, litter treatment,
season, WFPS, litter input rate, litter C input rate, litter N input rate, soil temperature and air



temperature as independent variables. The analysis was also repeated using ANCOVA on
Box-Cox transformed data. Both analyses revealed no significant statistical relationship
between $N_2O$ flux and any of these environmental variables, with the exception of soil
temperature, which showed only a weak positive relationship to $N_2O$ flux when the data was
analysed using the GLM ($P < 0.05$). This relationship was not detected using ANCOVA.
Bivariate regression of soil temperature against $N_2O$ flux indicated that the relationship was
relatively weak, with $r^2 = 0.01$ ($P < 0.05$).

**5.6 Nitrate addition experiment**
$^{15}N$-$N_2O$ and $^{15}N$-$N_2$ fluxes showed a biphasic response (Limmer and Steele, 1982), with
significantly different flux rates in the first 24 hours of incubation compared to the later
period of incubation (i.e. >24 hours onwards). Flux of $^{15}N$-$N_2O$, and $^{15}N$-$N_2$ were therefore
divided into early (≤24 hours) and late (>24 hours) phase flux.

**5.6.1 $^{15}N$-$N_2O$ flux**
For the $^{15}N$-$N_2O$ flux data, we used a full factorial ANOVA on Box-Cox transformed data with
habitat, N addition level, incubation phase, and all their interaction terms as independent
variables. Habitat, incubation phase, and a habitat by incubation phase interaction all
significantly influenced flux, while N addition level and all other interaction terms were not
statistically significant (ANOVA, $F_{29, 149} = 5.66$, $P < 0.0001$; Figure 5). For habitat, upper
montane forest organic layer soils showed the highest flux (238 ± 160 ng $N_2O$-$^{15}N$ $g^{-1}$ $d^{-1}$)
(Fisher's LSD, $P < 0.05$). This was followed by lower montane (179 ± 48 ng $N_2O$-$^{15}N$ $g^{-1}$ $d^{-1}$)
and premontane (86 ± 16 ng $N_2O$-$^{15}N$ $g^{-1}$ $d^{-1}$) forest, which collectively showed intermediate
flux (Fisher's LSD, $P < 0.05$). Last, the lowest flux was observed for montane grasslands (11 ±
4 ng $N_2O$-$^{15}N$ $g^{-1}$ $d^{-1}$), followed by upper montane forest mineral layer soils (0.06 ± 0.01 ng
$N_2O$-$^{15}N$ $g^{-1}$ $d^{-1}$) (Fisher's LSD, $P < 0.05$). The high rate of flux attributed to the upper montane
forest organic layer soils was due to a strong effect of phase, with significant increase in flux
during the late phase of the incubation (Figure 5). For the incubation phase effect, late phase
flux was significantly greater than early phase flux (164 ± 66 ng $N_2O$-$^{15}N$ $g^{-1}$ $d^{-1}$ versus 42 ± 11
ng $N_2O$-$^{15}N$ $g^{-1}$ $d^{-1}$; t-Test, $P < 0.05$; Figure 5).





For the habitat by incubation phase interaction, further investigation revealed that this
relationship arose from the fact that different habitats varied in their flux during early and
late incubation phases (Figure 5). For example, during the early phase, lower montane and
premontane forests collectively showed the highest flux (Figure 5; 133 ± 46 and 64 ± 19 ng
$N_2O$-$^{15}N$ $g^{-1}$ $d^{-1}$, respectively) (Fisher's LSD, $P < 0.05$). Upper montane forest organic layer
soils and montane grassland soils collectively showed intermediate rates of flux (Figure 5; 8 ±
2 and 4 ± 1 ng $N_2O$-$^{15}N$ $g^{-1}$ $d^{-1}$, respectively), while upper montane forest mineral layer soils
showed the lowest flux (Figure 5; 0.04 ± 0.01 ng $N_2O$-$^{15}N$ $g^{-1}$ $d^{-1}$) (Fisher's LSD, $P < 0.05$). In
contrast, during the late phase, upper montane forest organic layer soils, lower montane
forest, and premontane forest now collectively showed the highest flux (469 ± 313 ng $N_2O$-
$^{15}N$ $g^{-1}$ $d^{-1}$, 224 ± 85 ng $N_2O$-$^{15}N$ $g^{-1}$ $d^{-1}$, and 108 ± 25 ng $N_2O$-$^{15}N$ $g^{-1}$ $d^{-1}$, respectively). The
lowest flux was from montane grasslands (18 ± 7 ng $N_2O$-$^{15}N$ $g^{-1}$ $d^{-1}$), followed by upper
montane forest mineral layer soils (0.08 ± 0.02 ng $N_2O$-$^{15}N$ $g^{-1}$ $d^{-1}$) (Fisher's LSD, $P < 0.05$).

**5.6.2 $^{15}N$-$N_2$ flux**
For the $^{15}N$-$N_2$ flux data, we used a full factorial ANOVA on Box-Cox transformed data with
habitat, N addition level, incubation phase, and all their interaction terms as independent
variables. Only habitat significantly influenced flux (Figure 5), while other terms were not
significant (ANOVA, $F_{29, 149}$ = 1.66, $P < 0.05$). Lower montane and upper montane forest
organic layer soils showed the highest flux (472 ± 139 and 576 ± 117 ng $N_2$-$^{15}N$ $g^{-1}$ $d^{-1}$,
respectively), while all other habitats showed similar flux rates (105 ± 19 ng $N_2$-$^{15}N$ $g^{-1}$ $d^{-1}$)
(Fisher's LSD, $P < 0.05$; Figure 5).

**5.6.3 $N_2O$ Yield**
For the $N_2O$ yield, we used a full factorial ANOVA on Box-Cox transformed data with habitat,
N addition level, incubation phase (i.e. early versus late), and all their interaction terms as
independent variables. We found that none of these factors predicted $N_2O$ yield (ANOVA,
$F_{29, 149}$ = 0.75, $P > 0.82$). The overall mean $N_2O$ yield for the pooled dataset was 0.53 ± 0.04.


**6. Discussion**
**6.1 Multi-annual trends in $N_2O$ flux among habitats and between seasons**



Montane forest and grassland ecosystems in the Kosñipata Valley were net sources of
atmospheric $N_2O$, affirming our prior results (Teh et al., 2014). The flux for this multi-annual
dataset were comparable to the preliminary values reported in our earlier publication, with
mean flux of $0.27 \pm 0.07$ mg N-$N_2O$ m$^{-2}$ d$^{-1}$ observed here over a 30 month period, compared
with $0.22 \pm 0.12$ mg N-$N_2O$ m$^{-2}$ d$^{-1}$ recorded over 13 months (Teh et al., 2014). Consistent
with our earlier report, flux from our Peruvian transect were greater than those from a
comparable study site in Ecuador (Wolf et al., 2011), which we attributed to higher N
content in lower elevation soils in Peru (Teh et al., 2014). The elevational trends reported
earlier still hold true for this multi-annual dataset (Teh et al., 2014); namely, significantly
greater $N_2O$ flux from lower elevation habitats (premontane forest, lower montane forest)
compared to higher elevation ones (upper montane forest, montane grasslands) (Figure 2a).
More favourable environmental conditions at lower elevations may explain these trends
(e.g. higher N availability, warmer temperatures; see below for further details).

Nitrous oxide flux for the Kosñipata Valley varied between seasons, with significantly greater
flux during the dry season compared to the wet season (Teh et al., 2014). However, this
overall trend was strongly influenced by the behaviour of lower montane forest, which
showed pronounced seasonality in $N_2O$ flux, whereas the other habitats showed little or no
seasonal differences (Table 3). For premontane forest, upper montane forest, and montane
grassland, weak seasonality in $N_2O$ flux may reflect the fact that environmental variables did
not vary strongly between seasons (Table 3), challenging our first hypothesis (**H1**). Instead,
environmental variables tended to vary more strongly among habitats (section 5.2). Analysis
of the environmental data repeatedly demonstrated that habitat accounted for the largest
proportion of variance in ANOVA models, with season accounting for a substantially smaller
proportion of the variance or none at all. Moreover, in cases where environmental variables
differed significantly between seasons, the actual numerical differences were often
relatively slight (Table 3). For example, while WFPS varied significantly between seasons, the
numerical difference in WFPS between dry season and wet season was 7.4 % WFPS for the
pooled data; i.e. $52.1 \pm 2.4$ versus $59.5 \pm 1.6$ % WFPS, respectively. Likewise, oxygen in the 0-
10 cm soil depth varied by less than 1 %, with a mean dry season value of $17.8 \pm 0.3$ %
compared to a wet season value of $16.8 \pm 0.4$ %. Soil temperature varied by less than 1.2 °C,
with a mean dry season value of $13.9 \pm 0.4$ °C compared to a wet season value of $15.1 \pm 0.3$





°C. Other variables, such as air temperature and resin-extractable $NO_3^-$ did not vary
significantly between seasons at all.

Lower montane forest is the only habitat that showed evidence of seasonal fluctuations in
$N_2O$ flux driven by variability in environmental conditions. This is evidenced by the results of
multiple regression analysis of environmental variables against $N_2O$ flux (section 5.3). Key
variables found to influence $N_2O$ flux included air temperature, soil temperature, WFPS, and
resin-extractable $NH_4^+$ flux. According to the multiple regression analysis, the dominant
environmental regulator for $N_2O$ flux was air temperature, which showed a negative
relationship with $N_2O$ flux. While we are not entirely certain why air temperature was
negatively correlated with flux; one possible explanation is that this relationship reflects the
effect of air temperature on some other process linked to $N_2O$ flux, such as drying of surface
soil layers. Higher air temperatures may have led to increased evaporation in surface soil
horizons, reducing rates of N cycling. This is a phenomenon we have observed in other
warm, seasonally-dry environments (Teh et al., 2011), and we found limited evidence for this
interpretation of the data in the weak but statistically significant inverse relationship
between air temperature and WFPS ($r^2$ = 0.12, $P$ < 0.002; data not shown). The positive
relationship between soil temperature is perhaps more intuitive to interpret, and may
reflect enhanced microbial activity as the soil warms. Likewise, the negative relationship
with WFPS and $N_2O$ flux probably reflects enhanced $N_2O$ reductase activity and greater
denitrification to $N_2$ with increasingly anaerobic conditions (Morley and Baggs, 2010;Morley
et al., 2008). Last, the inverse relationship between resin-extractable $NH_4^+$ and $N_2O$ flux may
reflect competition for $NO_3^-$ between denitrification and dissimilatory nitrate reduction to
ammonium (DNRA), the two nitrate-reducing processes that are believed to be relatively
common in wet, organic matter-rich tropical soils (Silver et al., 2001). Of course, one puzzling
feature of this data is the divergent relationships that air temperature and soil temperature
show with $N_2O$ flux. We believe that the most likely explanation for this is that these two
environmental variables are, to some extent, decoupled from each other in these montane
habitats, leading to the two variables behaving differently from each other and acting as
least quasi-independently on $N_2O$ flux. This is evidenced by the weak positive correlation
between air and soil temperature in lower montane forest ($r^2$ = 0.20, $P$ < 0.0001), which
suggests that a large proportion of the variance in soil temperatures (i.e. up to 80 %) are





explained by other environmental factors, and not by ambient air temperature alone.
However, it is important to note that interpretation of these results must be treated with
some caution, given that the model as a whole was only on the borderline of statistical
significance ($P < 0.07$, $r^2 = 0.36$).

One other important difference between this publication and our earlier work is that
topography no longer appears to be an important driving variable in this multi-annual
dataset. While the basin landform showed significantly lower $N_2O$ flux than the other
landforms when the effect of topography was investigated in isolation, a more
comprehensive statistical analysis, which included topography and other variables (e.g.
habitat, season, environmental conditions), suggests that topography is not a significant
predictor of $N_2O$ flux. Instead, the effects of topography may be contingent upon or co-vary
with habitat, rather than acting independently of it.

**6.2 Effects of soil moisture on $N_2O$ flux**
Results from our laboratory-based WFPS manipulations suggest that soil moisture content
plays a significant role in modulating $N_2O$ flux. This finding is noteworthy because our prior
research suggested that there was no direct relationship between $N_2O$ flux and WFPS (Teh et
al., 2014), and challenged our broader theoretical understanding of the role that soil
moisture plays in regulating $N_2O$ flux (Firestone and Davidson, 1989;Firestone et al.,
1980;Weier et al., 1993). However, the response of $^{15}N$-$N_2O$ flux and other response
variables (e.g. $^{15}N$-$N_2$ flux, $N_2O$ yield) were complex and non-linear, falsifying our second
hypothesis (**H2**). Rather than $^{15}N$-$N_2O$ flux increasing progressively with WFPS, as predicted
by **H2** and denitrification theory (Firestone and Davidson, 1989;Firestone et al., 1980;Weier
et al., 1993), we observed two distinct and separate peaks in $^{15}N$-$N_2O$ flux. The highest $^{15}N$-
$N_2O$ flux was observed in the 90 and 50 % WFPS treatments, while the 30 and 70 % WFPS
treatments showed significantly lower flux (Fisher's LSD, $P < 0.05$; Figure 4). This unexpected
result may reflect competition for substrates (e.g. $NO_3^-$, labile organic C) among nitrate-
reducing processes such as denitrification and DNRA (Silver et al., 2001), or may indicate that
$N_2O$ is being produced from DNRA (Streminska et al., 2012).



$^{15}N-N_2$ flux and $N_2O$ yield also showed intriguing and unexpected trends. For example, $^{15}N-N_2$
flux was highest flux in the 90 % WFPS treatment (Fisher's LSD, $P < 005$), but did not differ
significantly among the other treatments (Figure 4). Likewise, $N_2O$ yield was highest in the 70
% WFPS treatment (0.51 ± 0.06), above and below which significantly smaller proportions of
$^{15}N$ were emitted as $N_2O$ (Fisher's LSD, $P < 0.05$). These results are surprising because
denitrification theory predicts that decreases in WFPS should lead to a reduction in $N_2$ flux
and increases in $N_2O$ yield (Firestone and Davidson, 1989;Firestone et al., 1980;Weier et al.,
1993), as $N_2O$ reductase is increasingly suppressed by drier and more oxic soil conditions
(Burgin and Groffman, 2012;Weier et al., 1993;Firestone et al., 1980;Morley and Baggs,
2010;Morley et al., 2008). One explanation for this is that $N_2O$ production under drier
conditions (i.e. <50 % WFPS) may be occurring in anaerobic microsites (Keller et al.,
1993;Silver et al., 1999).

**6.3 $N_2O$ flux not constrained by labile organic matter availability**
Nitrous oxide flux was unaffected by variations in leaf litter-fall, partially challenging our
third hypothesis (**H3**). This finding runs counter to the results from lowland tropical forests
(Sayer et al., 2011), where trace gas flux can be strongly influenced by changes in labile
organic matter inputs, such as leaf litter. The relative insensitivity of these montane
ecosystems to changes in leaf litter-fall, a proxy for labile organic matter inputs, may be due
to the relatively large size of soil organic matter pools in these soils (Zimmermann et al.,
2012, Zimmermann et al., 2009a, Zimmermann et al., 2010b), which could buffer $N_2O$
production against short-term fluctuations in labile organic matter availability. Moreover,
because of the relatively large soil organic matter stocks,  and $N_2O$ emission could be more
strongly constrained by other factors, such as N availability, soil WFPS or pH. This finding is
significant for understanding and modelling process-based controls on $N_2O$ flux, as many
bottom-up, process-based models assume that N cycling and turnover of labile organic
matter are linked through processes such as litter production and decomposition (Li et al.,
2000;Werner et al., 2007). While not disproving these assumptions, these data suggest that
the linkage between litter production and $N_2O$ flux are weak in these montane
environments.

**6.4 Importance of $NO_3^-$ in regulating $N_2O$ flux**





One of the principal hypotheses raised by our earlier research is that $N_2O$ flux is strongly
limited by $NO_3^-$ across this tropical elevation gradient (Teh et al., 2014). The detailed,
process-oriented studies conducted here provide evidence that supports this claim,
indicating that longer-term, time-averaged patterns in $NO_3^-$ availability among habitats
influence $N_2O$ flux. The strongest evidence comes from the $^{15}N$-$N_2O$ flux data from our $^{15}N$-
$NO_3^-$ addition experiment. Trends in $^{15}N$-$N_2O$ flux echoed patterns in our field data and prior
denitrification potential experiments (Teh et al., 2014). Namely, we observed an inverse
trend in $^{15}N$-$N_2O$ flux with elevation, with significantly higher $^{15}N$-$N_2O$ flux from lower
elevation premontane (86 ± 16 ng $N_2O$-$^{15}N$ $g^{-1}$ $d^{-1}$) and lower montane (179 ± 48 ng $N_2O$-$^{15}N$
$g^{-1}$ d-1) forests, compared to higher elevation upper montane forest mineral layer soils (0.06
± 0.01 ng $N_2O$-$^{15}N$ $g^{-1}$ $d^{-1}$) and montane grasslands (11 ± 4 ng $N_2O$-$^{15}N$ $g^{-1}$ $d^{-1}$) (Figure 5a). This
pattern in $^{15}N$-$N_2O$ flux follows trends in resin-extractable $NO_3^-$ flux, implying that $NO_3^-$ may
constrain the potential of these soil to emit $N_2O$ (Figure 2a-b, Figure 5a) (Teh et al., 2014).
The exception to this pattern is upper montane forest organic layer soils, which showed the
highest flux when incubated under laboratory conditions (Figure 5). However, it is important
to note that the significantly lower bulk density of the organic horizon in upper montane
forests (~0.06 g $cm^{-3}$ for the O horizon versus ~0.6 g $cm^{-3}$ for the mineral horizon) means that
this O layer makes a smaller proportional contribution to $N_2O$ flux than soils from lower
mineral horizons (Zimmermann et al., 2009a;Zimmermann et al., 2009b).

Furthermore, the behaviour of the $NO_3^-$ amended soils during the early (≤24 hours) and late
(>24 hours) phases of the incubation suggest that soils from more N-poor habitats showed a
greater proportional increase in $^{15}N$-$N_2O$ flux following $NO_3^-$ addition than N-rich habitats,
suggesting that $^{15}N$-$N_2O$ flux was more $NO_3^-$ limited in N-poor environments (Figure 5). For
example, soils from the upper montane forest organic layer, montane grasslands, and upper
montane forest mineral layer showed the lowest early phase $^{15}N$-$N_2O$ flux, but the greatest
proportional increase in flux during the late incubation phase, rising by a factor of 59, five,
and two, respectively. In contrast, lower montane and premontane forest soils, which
showed the highest $NO_3^-$ availability and $N_2O$ flux in the field, and the greatest early phase
$^{15}N$-$N_2O$ flux in the incubations, showed the smallest proportional increase in the late
incubation phase (i.e. 1.7 times increase). Overall, these data imply that $^{15}N$-$N_2O$ flux from N-



poor habitats are more strongly $NO_3^-$ limited, whereas $N_2O$ flux from more N-rich soils may
be more heavily constrained by other environmental factors.

The other field and laboratory data were more equivocal, reflecting the complex and
potentially confounding environmental controls on $N_2O$ flux (Groffman et al., 2009). For
example, while lower $N_2O$ flux was associated with more N-poor habitats, $N_2O$ flux was only
weakly correlated with resin-extractable $NO_3^-$ flux ($r^2$ = 0.03, $P$ <0.03). Moreover, for the
laboratory-based $NO_3^-$ addition experiment, we found no evidence that these soils
responded to short-term increases in $NO_3^-$ availability, at least within the concentration
range that we used in this experiment. $^{15}N$-$N_2O$ flux, $^{15}N$-$N_2$ flux, and $N_2O$ yield were not
directly influenced by the amount of $^{15}N$-$NO_3^-$ added (Figure 5). Rather, ANCOVA suggests
that $^{15}N$-$N_2O$ and $^{15}N$-$N_2$ fluxes were better-predicted by habitat. $N_2O$ yield, normally a
sensitive indicator of $NO_3^-$ availability (Blackmer and Bremner, 1978;Weier et al.,
1993;Parton et al., 1996), showed no immediate response to the amount of $^{15}N$-$NO_3^-$ added,
nor any of the other explanatory variables. One explanation for this, consistent with the
notion that $N_2O$ flux is $NO_3^-$ limited, is that nitrate-reducing microbes in these soils may have
a relatively low half-saturation constant ($K_m$) for $NO_3^-$, and effectively utilize $NO_3^-$ whenever
concentrations increase above background levels (Holtan-Hartwig et al., 2000). As a
consequence, we may be unable to differentiate among $NO_3^-$ treatments because the $NO_3^-$
addition levels that we used all exceeded the $K_m$ for in these soils. This finding is also
consistent with results from long-term N fertilization studies, which suggest that substantive
shifts in $N_2O$ flux are only likely to occur after prolonged exposure to high levels of N, rather
than due to transient fluctuations in N availability (Hall & Matson 1993; Koehler et al 2009;
Corre et al 2014).


**7. Conclusions**
Process-based studies of $N_2O$ flux from montane tropical ecosystems in the southern
Peruvian Andes affirms prior research suggesting that these ecosystems are potentially
important regional sources of $N_2O$ (Teh et al., 2014). Nitrous oxide flux originated primarily
from nitrate reduction rather than from nitrification, probably due to low pH soil conditions.
Contrary to our earlier research, we found only weak evidence for seasonal patterns in $N_2O$



flux, with the exception of lower montane forest, which showed significantly higher $N_2O$ flux
during the dry season compared to the wet season. Weak seasonal trends in $N_2O$ flux among
the other montane habitats probably stems from relatively modest variation in key
environmental drivers (e.g. temperature, WFPS, $NO_3^-$) between seasons. Nitrous oxide flux
was significantly influenced by soil moisture content, but the effect of soil moisture content
on $N_2O$ flux was complex and non-linear. Nitrous oxide flux showed a bimodal response to
increasing soil moisture content, with peaks in $N_2O$ flux at 90 and 50 % WFPS. These data
suggest that the effects of water on $N_2O$ flux are complicated by other factors, such as
competition for substrates among different nitrate-reducing processes, or shifts in the
amount of $N_2O$ derived from denitrification or DNRA. Substrate manipulation experiments
indicated that $N_2O$ flux was limited by $NO_3^-$, but unconstrained by the input rate of labile
organic matter (i.e. leaf litter). Nitrous oxide flux was relatively insensitive to short-term
variations in $NO_3^-$, and was better-predicted by longer-term, time-averaged variations in
$NO_3^-$ availability.


**8. Author Contributions**
TD designed the field and laboratory experiments, collected the field data, conducted the
laboratory experiments, processed the samples, analysed the data, and contributed to the
preparation of the manuscript. NJM contributed to the design of the laboratory
experiments, assisted in the sample analysis, assisted in the analysis of the laboratory data,
and contributed to the preparation of the manuscript. AJC and LPHQ assisted in the
collection of the field data and processing of the field samples. EMB, PM, MR, and PS
contributed to the experimental design and the preparation of the manuscript. YAT directed
the research, contributed to the design of the experiments, assisted in the analysis of the
field and laboratory data, and took the principal role in preparing the manuscript.


**9. Acknowledgements**
The authors would like to acknowledge the agencies that funded this research; the UK
Natural Environment Research Council (NERC; joint grant references NE/H006583,
NE/H007849 and NE/H006753). Patrick Meir was supported by an Australian Research





Council Fellowship (FT110100457). Javier Eduardo Silva Espejo, Walter Huaraca Huasco, and
the ABIDA NGO provided critical fieldwork and logistical support. Angus Calder (University of
St Andrews) and Vicky Munro (University of Aberdeen) provided invaluable laboratory
support. Thanks to Adrian Tejedor from the Amazon Conservation Association, who provided
assistance with site access and site selection at Hacienda Villa Carmen. This publication is a
contribution from the Scottish Alliance for Geoscience, Environment and Society
(http://www.sages.ac.uk).

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

**Table 1.** Site characteristics.

| Elevation Band m a.s.l. | Habitat | Latitude | Longitude | Mean Annual Temperature Soil 0-10 cm °C | Mean Annual Precipitation mm | Bulk density 0-10 cm g cm-3 | pH | Soil C:N 0-10 cm | Soil C 0-10 cm % | Clay 0-10 cm | Silt 0-10 cm | Sand 0-10 cm | Clay 10-30 cm | Silt 10-30 cm | Sand 10-30 cm | Landforms | Plots n | Flux Chambers n |
|---|---|---|---|---|---|---|---|---|---|---|---|---|---|---|---|---|---|---|
| | | | | | | | | | | | | Mineral Soil Particle Size | | | | | | |
| 500-1200 | Premontane forest | 12°53'43" | 71°23'04" | 20.5 | 5318 | 0.38 ± 0.03 (n = 21) | 3.4 ± 0.1 | 11.3 ± 0.2 | 7.9 ± 0.5 | 5.4 ± 0.3 | 68.8 ± 3.9 | 25.4 ± 15.9 | 8.9 ± 1.8 | 81.0 ± 1.7 | 10.3 ± 2.5 | ridge, slope, flat | 3 | 15 |
| 200-2200 | Lower montane forest | 13°2'56" | 71°32'13" | 17.2 | 2631 | 0.19 ± 0.03 (n = 17) | 3.4 ± 0.1 | 14.5 ± 0.2 | 25.2 ± 1.3 | 3.6 ± 0.4 | 67.3 ± 4.2 | 29.3 ± 4.5 | 7.2 ± 0.4 | 83.8 ± 0.8 | 9.0 ± 0.9 | ridge, slope, flat | 3 | 15 |
| 200-3200 | Upper montane forest | 13°11'24" | 71°35'13" | 10.7 | 1706 | 0.41 ± 0.02 (n = 12) | 3.9 ± 0.1 | 16.8 ± 0.4 | 16.3 ± 1.0 | 5.1 ± 0.9 | 57.1 ± 7.9 | 37.9 ± 8.7 | 4.4 ± 2.0 | 46.5 ± 16.2 | 49.1 ± 18.1 | ridge, slope | 3 | 15 |
| 200-3700 | Montane grassland | 13°07'19" | 71°36'54" | 9.3 | 2200 | 0.36 ± 0.03 (n = 27) | 4.1 ± 0.1 | 12.9 ± 0.4 | 16.0 ± 1.0 | 2.6 ± 0.2 | 54.4 ± 3.0 | 43.0 ± 3.2 | n/a | n/a | n/a | ridge, slope, flat, basin | 4 | 20 |



**Table 2.** Description of the water-filled pore space and $NO_3^-$ addition treatments for the
laboratory manipulation experiments.

| Habitat | Experimental Treatment | Soil Depth | Soil Type | WFPS % | Inorganic N added $\mu g\ N\ (g\ soil)^{-1}$ | $^{15}N$ Tracer | Replicate n |
|---|---|---|---|---|---|---|---|
| **WATER-FILLED PORE SPACE** | | | | | | | |
| Premontane forest | 90 % WFPS | 0-10 | mineral | 90 | 200 | $^{15}NH_4^{15}NO_3$ | 5 |
| | 90 % WFPS | 0-10 | mineral | 90 | 200 | $^{14}NH_4^{15}NO_3$ | 5 |
| | 70 % WFPS | 0-10 | mineral | 70 | 200 | $^{15}NH_4^{15}NO_3$ | 5 |
| | 70 % WFPS | 0-10 | mineral | 70 | 200 | $^{14}NH_4^{15}NO_3$ | 5 |
| | 50 % WFPS | 0-10 | mineral | 50 | 200 | $^{15}NH_4^{15}NO_3$ | 5 |
| | 50 % WFPS | 0-10 | mineral | 50 | 200 | $^{14}NH_4^{15}NO_3$ | 5 |
| | 30 % WFPS | 0-10 | mineral | 30 | 200 | $^{15}NH_4^{15}NO_3$ | 5 |
| | 30 % WFPS | 0-10 | mineral | 30 | 200 | $^{14}NH_4^{15}NO_3$ | 5 |
| Lower montane forest | 90 % WFPS | 0-10 | mineral | 90 | 200 | $^{15}NH_4^{15}NO_3$ | 5 |
| | 90 % WFPS | 0-10 | mineral | 90 | 200 | $^{14}NH_4^{15}NO_3$ | 5 |
| | 70 % WFPS | 0-10 | mineral | 70 | 200 | $^{15}NH_4^{15}NO_3$ | 5 |
| | 70 % WFPS | 0-10 | mineral | 70 | 200 | $^{14}NH_4^{15}NO_3$ | 5 |
| | 50 % WFPS | 0-10 | mineral | 50 | 200 | $^{15}NH_4^{15}NO_3$ | 5 |
| | 50 % WFPS | 0-10 | mineral | 50 | 200 | $^{14}NH_4^{15}NO_3$ | 5 |
| | 30 % WFPS | 0-10 | mineral | 30 | 200 | $^{15}NH_4^{15}NO_3$ | 5 |
| | 30 % WFPS | 0-10 | mineral | 30 | 200 | $^{14}NH_4^{15}NO_3$ | 5 |
| Upper montane forest | 90 % WFPS | 10-20 | mineral | 90 | 20 | $^{15}NH_4^{15}NO_3$ | 5 |
| | 90 % WFPS | 10-20 | mineral | 90 | 20 | $^{14}NH_4^{15}NO_3$ | 5 |
| | 70 % WFPS | 10-20 | mineral | 70 | 20 | $^{15}NH_4^{15}NO_3$ | 5 |
| | 70 % WFPS | 10-20 | mineral | 70 | 20 | $^{14}NH_4^{15}NO_3$ | 5 |
| | 50 % WFPS | 10-20 | mineral | 50 | 20 | $^{15}NH_4^{15}NO_3$ | 5 |
| | 50 % WFPS | 10-20 | mineral | 50 | 20 | $^{14}NH_4^{15}NO_3$ | 5 |
| | 30 % WFPS | 10-20 | mineral | 30 | 20 | $^{15}NH_4^{15}NO_3$ | 5 |
| | 30 % WFPS | 10-20 | mineral | 30 | 20 | $^{14}NH_4^{15}NO_3$ | 5 |
| Montane grassland | 90 % WFPS | 0-10 | mineral | 90 | 20 | $^{15}NH_4^{15}NO_3$ | 5 |
| | 90 % WFPS | 0-10 | mineral | 90 | 20 | $^{14}NH_4^{15}NO_3$ | 5 |
| | 70 % WFPS | 0-10 | mineral | 70 | 20 | $^{15}NH_4^{15}NO_3$ | 5 |
| | 70 % WFPS | 0-10 | mineral | 70 | 20 | $^{14}NH_4^{15}NO_3$ | 5 |
| | 50 % WFPS | 0-10 | mineral | 50 | 20 | $^{15}NH_4^{15}NO_3$ | 5 |
| | 50 % WFPS | 0-10 | mineral | 50 | 20 | $^{14}NH_4^{15}NO_3$ | 5 |
| | 30 % WFPS | 0-10 | mineral | 30 | 20 | $^{15}NH_4^{15}NO_3$ | 5 |
| | 30 % WFPS | 0-10 | mineral | 30 | 20 | $^{14}NH_4^{15}NO_3$ | 5 |
| **NITRATE ADDITION** | | | | | | | |
| Premontane forest | control | 0-10 | mineral | 80 | n/a | n/a | 5 |
| | +50 % background $NO_3^-$ | 0-10 | mineral | 80 | 78 ± 6 | $K^{15}NO_3$ | 5 |
| | +100 % background $NO_3^-$ | 0-10 | mineral | 80 | 157 ± 12 | $K^{15}NO_3$ | 5 |
| | +150 % background $NO_3^-$ | 0-10 | mineral | 80 | 235 ± 17 | $K^{15}NO_3$ | 5 |
| Lower montane forest | control | 0-10 | mineral | 80 | n/a | n/a | 5 |
| | +50 % background $NO_3^-$ | 0-10 | mineral | 80 | 70 ± 6 | $K^{15}NO_3$ | 5 |
| | +100 % background $NO_3^-$ | 0-10 | mineral | 80 | 140 ± 12 | $K^{15}NO_3$ | 5 |
| | +150 % background $NO_3^-$ | 0-10 | mineral | 80 | 210 ± 18 | $K^{15}NO_3$ | 5 |
| Upper montane forest | control | 0-10 | organic | 80 | n/a | n/a | 5 |
| | +50 % background $NO_3^-$ | 0-10 | organic | 80 | 9 ± 2 | $K^{15}NO_3$ | 5 |
| | +100 % background $NO_3^-$ | 0-10 | organic | 80 | 18 ± 5 | $K^{15}NO_3$ | 5 |
| | +150 % background $NO_3^-$ | 0-10 | organic | 80 | 27 ± 7 | $K^{15}NO_3$ | 5 |
| | control | 10-20 | mineral | 80 | n/a | n/a | 5 |
| | +50 % background $NO_3^-$ | 10-20 | mineral | 80 | 9 ± 4 | $K^{15}NO_3$ | 5 |
| | +100 % background $NO_3^-$ | 10-20 | mineral | 80 | 19 ± 7 | $K^{15}NO_3$ | 5 |
| | +150 % background $NO_3^-$ | 10-20 | mineral | 80 | 28 ± 11 | $K^{15}NO_3$ | 5 |
| Montane grassland | control | 0-10 | mineral | 80 | n/a | n/a | 5 |
| | +50 % background $NO_3^-$ | 0-10 | mineral | 80 | 3 ± 1 | $K^{15}NO_3$ | 5 |
| | +100 % background $NO_3^-$ | 0-10 | mineral | 80 | 6 ± 2 | $K^{15}NO_3$ | 5 |
| | +150 % background $NO_3^-$ | 0-10 | mineral | 80 | 9 ± 4 | $K^{15}NO_3$ | 5 |



**Table 3.** Net N$_2$O flux and abiotic environmental variables for each habitat for the wet and dry season. Lower case letters indicate difference among seasons within habitats (*t*-Test on Box-Cox transformed data, *P* < 0.05). Values reported here are means and standard errors.

| Habitat | N$_2$O mg N-N$_2$O m$^{-2}$ d$^{-1}$ | | WFPS % | | Soil Temperature °C | | Air Temperature °C | | Oxygen % | | NO$_3^-$ µg N-NO$_3^-$ (g resin)$^{-1}$ d$^{-1}$ | | NH$_4^+$ µg N-NH$_4^+$ (g resin)$^{-1}$ d$^{-1}$ | |
|---|---|---|---|---|---|---|---|---|---|---|---|---|---|---|
| | Wet Season | Dry Season | Wet Season | Dry Season | Wet Season | Dry Season | Wet Season | Dry Season | Wet Season | Dry Season | Wet Season | Dry Season | Wet Season | Dry Season |
| Premontane | 0.71 ± 0.25 a n = 130 | 0.79 ± 0.26 a n = 98 | 51.9 ± 1.6 a n = 135 | 51.2 ± 2.1 a n = 135 | 20.7 ± 0.1 a n = 143 | 20.2 ± 0.1 b n = 120 | 21.5 ± 0.3 n = 143 | 20.4 ± 0.5 n = 120 | 19.4 ± 0.2 a n = 52 | 19.6 ± 0.2 a n = 36 | 23.2 ± 3.6 a n = 89 | 22.1 ± 2.1 a n = 96 | 31.4 ± 13.0 n = 90 | 11.3 ± 1.8 n = 95 |
| Lower montane | 0.09 ± 0.08 a n = 212 | 1.02 ± 0.58 b n = 137 | 42.2 ± 1.0 a n = 271 | 34.0 ± 1.4 b n = 179 | 18.1 ± 0.1 a n = 254 | 17.3 ± 0.2 b n = 164 | 18.9 ± 0.3 n = 254 | 18.3 ± 0.2 n = 164 | 19.2 ± 0.2 a n = 146 | 19.2 ± 0.1 a n = 81 | 11.8 ± 1.9 a n = 123 | 7.8 ± 1.4 a n = 94 | 20.2 ± 5.4 n = 124 | 8.6 ± 0.9 n = 93 |
| Upper montane | 0.06 ± 0.09 a n = 207 | 0.01 ± 0.11 a n = 146 | 42.0 ± 1.3 a n = 264 | 24.3 ± 1.4 b n = 180 | 11.8 ± 0.1 a n = 255 | 10.9 ± 0.2 b n = 165 | 12.8 ± 0.2 n = 255 | 12.5 ± 0.3 n = 165 | 18.7 ± 0.2 a n = 165 | 18.5 ± 0.2 a n = 109 | 1.4 ± 0.2 a n = 128 | 0.6 ± 0.2 b n = 91 | 22.5 ± 6.3 n = 129 | 11.3 ± 1.4 n = 93 |
| Montane grassland | -0.01 ± 0.11 a n = 238 | 0.19 ± 0.12 a n = 160 | 88.5 ± 0.3 a n = 303 | 88.3 ± 0.5 a n = 184 | 11.6 ± 0.1 a n = 282 | 9.0 ± 0.2 b n = 205 | 11.4 ± 0.3 n = 284 | 12.0 ± 0.5 n = 205 | 12.2 ± 0.9 a n = 176 | 15.4 ± 0.8 b n = 117 | 1.5 ± 0.4 a n = 128 | 2.1 ± 0.4 a n = 81 | 17.8 ± 4.3 n = 135 | 7.2 ± 0.8 n = 84 |



**Figure 1.** Map of study sites across the Kosñipata Valley, Manu National Park, Peru.



**Figure 2.** Plot-averaged (a) net $N_2O$ flux, (b) water-filled pore space, and (c) resin-extractable $NO_3^-$ flux among habitats. Boxes enclose the interquartile range, whiskers indicate the 90th and 10th percentiles. Lower case letters indicate statistically significant differences among means (Fisher's LSD, $P < 0.05$).

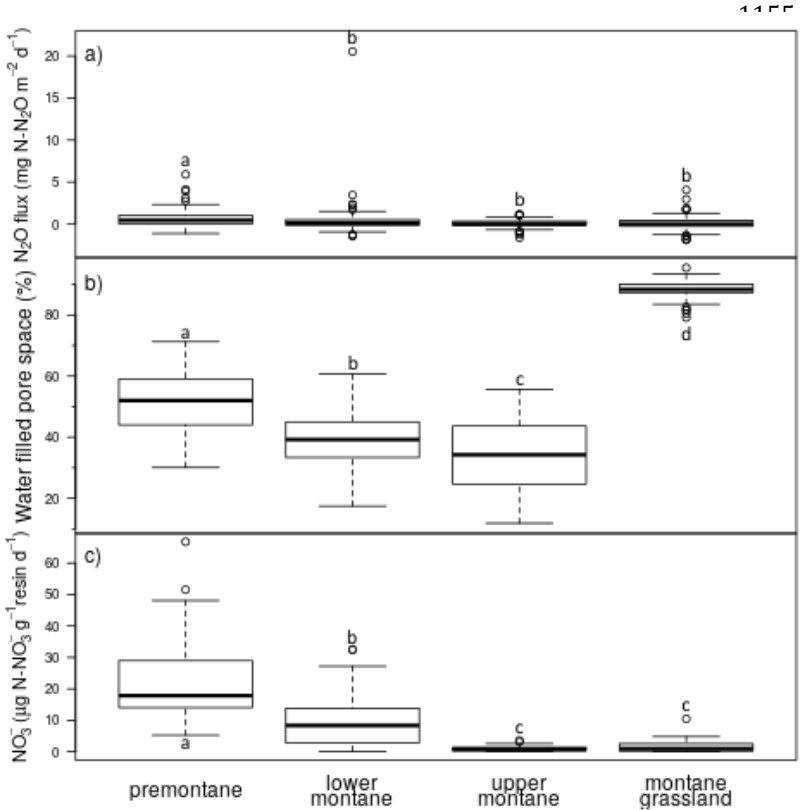





**Figure 3.** Time series of net N$_2$O flux and water-filled pore space (WFPS) for the whole data. Panels indicate data for (a) premontane forest, (b) lower montane forest, (c) upper montane forest, and (d) montane grasslands for the 30-month study period beginning in January 2011 and ending in June 2013. The broken horizontal line running across each panel denotes the overall mean N$_2$O flux or WFPS for that habitat. The broken line in each box indicate median values and the black lines indicate means. Dry and wet seasons are denoted by vertical shading on the graph, with the dry season (May to September) identified in white and the wet season (October to April) in light blue.

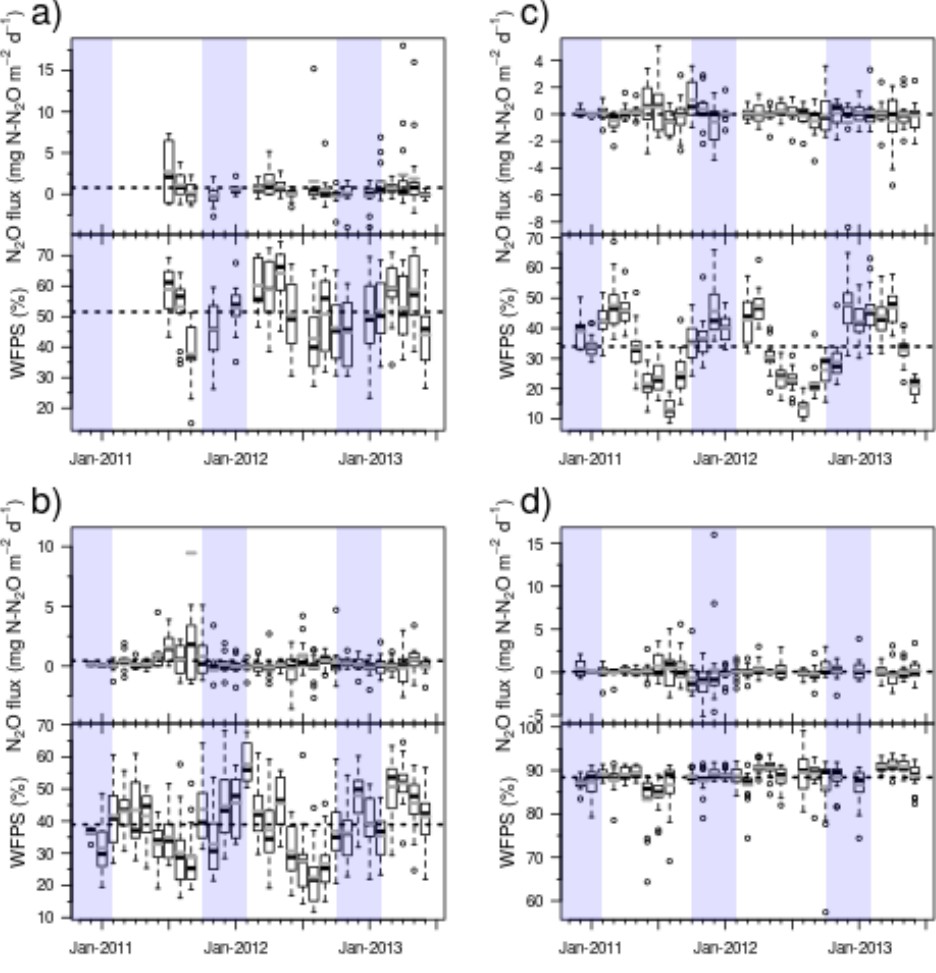





**Figure 4.** Total (a) $^{15}$N-N$_2$O flux and (b) $^{15}$N-N$_2$ flux during the early (≤24 hours) and late (>24
hours) incubation phases of the water-filled pore space (WFPS) experiment. Results from the
90 % WFPS treatment are shown in dark-grey, while data from the 70 %, 50 %, and 30 %
treatments are shown in mid-grey, light-grey, and white, respectively. The bar charts show
means and standard errors.

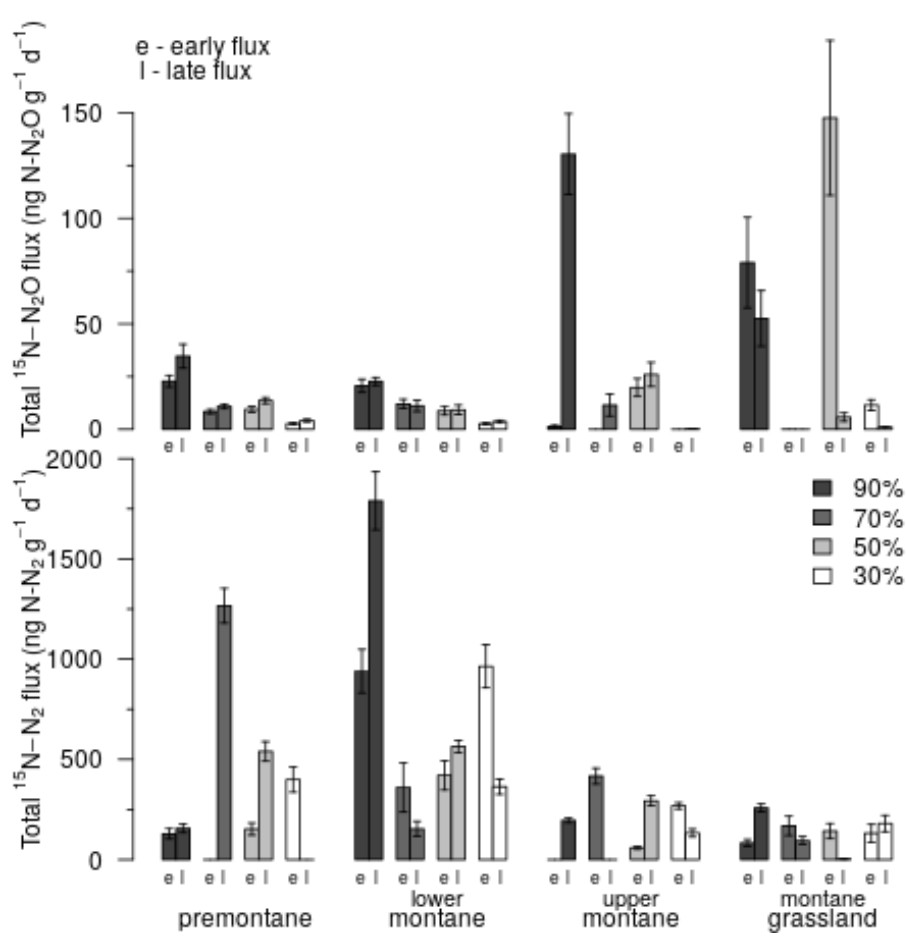






**Figure 5.** (a) $^{15}$N-N$_2$O flux and (b) $^{15}$N-N$_2$ flux during the early (≤24 hours) and late (>24 hours)
incubation phases of the NO$_3^-$ addition experiment. Results from the +50 % NO$_3^-$ addition are
shown in dark-grey, while data from the +100 % and +150 % treatments are shown in mid-
grey and light-grey, respectively. The bar charts show means and standard errors.

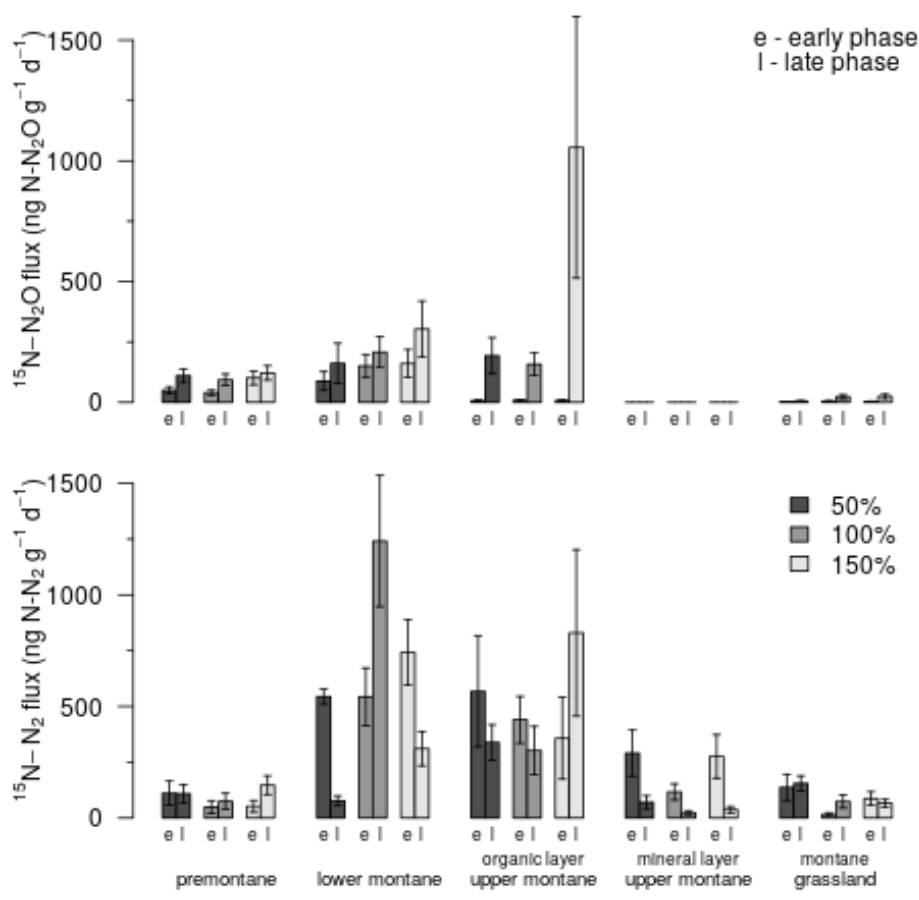
