# Peer review of "1. Title page: 2 Complex controls on nitrous oxide flux across a long elevation gradient in the tropical 3 4 Peruvian Andes 5 Torsten Diem1,2, Nicholas J. Morley1, Adan Julian Ccahuana3, Lidia Priscila Huaraca Quispe3, 6 Elizabeth M. Baggs4</sup"

_Biogeosciences, 2017_

## Referee Comment (RC1) · Anonymous Referee #1 · 3 May 2017

In 2014, some of the authors of the present publication published in Biogeosciences (doi:10.5194/bg-11-2325-2014) a paper entitled "Methane and nitrous oxide fluxes across an elevation gradient in the tropical Peruvian Andes". It was a very interesting paper because there is only little information about soil nitrous oxide fluxes and their controls in tropical montane forest soils. In their one-year study they pointed out that nitrous oxide fluxes were primarily driven by denitrification and that nitrate availability was the principal constraint on soil nitrous oxide fluxes followed by soil moisture. In the present study Diem and colleagues extended their time-series to multi-annual time scales to identify controls of longer-term climatic variability, soil moisture and substrate availability on nitrous oxide fluxes in greater detail. They found out that habitat/elevation

site, a proxy for nitrate availability under field conditions, was the best predictor for nitrous oxide fluxes. It is a great study. I have only few suggestions.

Major suggestions:

I would suggest to reformulate the introduction and the hypotheses. The main message is that habitat/elevation – a proxy for NO3 availability in the field – is the best predictor for N2O flux and that seasonal differences of N2O flux and environmental variables were most pronounced at the lower montane forest site, where N2O flux was best explained by a combination of temperature, WFPS and N-availability. I would remove substrate availability and/or labile organic matter because it does not enrich the discussion but rather blur the main message. I think it is sufficient to discuss an absent correlation between N2O flux and variations in leaf-litter fall in one or two sentences and not in a whole discussion section (L827-L843).

At the moment it seems that results and discussion section are dominated by the description and interpretation of the experimental results in the lab. I am very sceptical whether the results from the laboratory-based nitrogen and WFPS manipulations can be directly linked to the results obtained in the field, especially when they are as puzzling and surprising as in the present study (i.e. WFPS-manipulation study). Substrate availability, nutrient limitations and a cascade of active microbial community composition may have drastically changed during transportation from the field site in Peru to Aberdeen. As long as there is no clearer picture about the active microbial community in the samples before and after transport, all of the nutrient and trace gas flux observations during incubation experiments have only potential implications. Additionally, the ratio of N2O to N2 production is pH-dependent. Did you check for potential pH changes upon transportation?

What I find more fascinating is the observation of a negative relationship between WFPS and N2O flux in the field. The authors suggest that increasingly anaerobic conditions may stimulate N2O reductase activity and lead to greater denitrification to N2.

[Figure]

This strengthens the assumption of Mueller et al. 2015 who suggested that gaseous N loss was likely dominated by N2 rather than N2O in Ecuadorian montane forest soils. Taken together, this finding may be generalized to tropical montane forest ecosystems.

This leads me to another suggestion. Many parts of the discussion section read like a repetition or better description of the results section (e.g. L740-L760; L814-L818; L851-L858; L869-L876; L881-L891). Moreover, the links between different parts are laborious (e.g. L730-L734; L751-L755; L784-790; L880). I think it is necessary to make the reading more "fluid". Many sentences in the results and discussion section begin with "For example" (e.g. L534, L620, L689, L745, L814). I think the discussion section would benefit if present results would be more interpreted in the light of recent publications (e.g. Baldos et al. 2015; Mueller et al. 2015; Nottingham et al. 2015).

Minor comments:

L45-L48: This should also be mentioned in the conclusion section L98: . . .derived from (missing word) L290: What is the sampling size of the background concentration measurements? L300: What was the length of time between sampling and analysis? L827-L843: Remove heading and shorten section. L880-L900: Does this section really enrich the discussion? L906-L907: "Nitrous oxide flux originated primarily from nitrate reduction rather than from nitrification, probably due to low pH soil condition". Influence of pH has not been discussed in previous sections. L912: It should be clearly stated whether results were obtained from incubation experiments or from the field. Table1, Figure 3: Table and figure are very difficult to read. May be you can upload tables and figures in a higher resolution.

References: Baldos et al. 2015 (DOI: 10.1890/14-0295.1) Mueller et al. 2015 (DOI: 10.3389/feart.2015.00066) Nottingham et al. (DOI:10.5194/bg-12-6071-2015)

---

## Referee Comment (RC2) · Anonymous Referee #2 · 5 May 2017

The authors address the complex issue of N2O emissions that is globally, even more for tropical forests, and particularly for montane tropical forests widely unconstrained. The experimental setup in the field and in the laboratory were designed to capture mechanisms that affect N2O production and emissions. These effects include soil moisture, substrate availability (both mineral nitrogen and labile organic matter), soil moisture, oxygen, and temperature. They further analyzed more indirect predictors such as biome type, topography, seasonality, year to year variability - as well as interacting effects among these potential drivers for N2O production. The major outcome of this study is that the controls on N2O emissions remain elusive and in parts counter existing knowledge. In particular, the study finds little seasonal variability despite strong

seasonality in wetness. Further, soil moisture experiments suggest not the straight-forward controls as they are being used in conceptual and numerical models. The exhaustive work done in soils in difficult and previously unsampled environment, as well as (in my view) important laboratory experiments that complement the field work. The data deserves dissemination to the scientific public. However, I do have some suggestions and comments on the presentation and interpretation of the data.

Organization: The sheer number of observations and experiments, the exhaustive statistical analysis makes, and the resulting (complex pattern) makes it hard to write a clean story. Yet I think the authors should give the presentation some more thought. The result section is full of statistical test results, I am wondering if the tests applied and their results would not be better confined to tables, while the result text focuses more on the most important patterns.

Hypotheses: I would love to see a bit more nuanced hypotheses: Teh et al., 2014 already show an "odd" relationship with soil moisture (i.e. unexpected highs during dry season compared to wet season). Could better hypotheses be developed based on this earlier data? In light of previous work done at the site, H1 and H2 are fairly generic. Similarly, since the paper also addresses elevation gradients (or transitions from premontane tropical forests to montane grasslands, perhaps there are potential to use that gradient to set up additional hypotheses (What are expectations if compared to [seasonally dry] lowland tropical systems?).

Seasonality: Looking at the time series, it seems to me from the get go there is no direct seasonal effect. However, there are curious seasonal patterns: Soil moisture seems to lag quite a bit the precipitation (i.e. soil moisture seems to increase at the beginning of the dry season before it diminishes, while soil moisture continues to decline after the onset of the wet season). Much harder to discern, but just eyeballing the data in Fig 3, it seems there is a seasonal pattern of N2O emissions that it out of phase with seasonality, and is also out of phase with soil moisture. I do not have a mechanistic explanation how such lags can be formed given that often the first rain

leads to strong pulses in denitrification. Nor do I know whether the patterns I seem to recognize are really there if further scrutinized. Yet I am wondering if there should be some exploration with the inclusion of lag in the analysis. Perhaps the authors toyed with it and did not pan out, However, I would be curious to know either way.

Bimodal soil moisture response: The authors put strong emphasis on the bimodal soil moisture response of N2O emissions with peaks at 90 % and 50 % water filled pore space – stating it both in the abstract and the conclusion. However, this is in my view not clearcut, occurring only in some of the sampled soils. The results and the discussion acknowledge this. Is there a way to nuance the abstract and conclusion, such that the result do not come over as overstated?

Gradient nitrogen-rich -> nitrogen poor. In several places there is mention that the premontane and the lower montane habitats are nitrogen rich, whereas the higher elevations are considered nitrogen poor. It is perhaps worthwile to define N rich and N poor explicitly (for example by resin bag mineral N). This seems to be very important, given that nitrate availability may be a strong driver for N2O production.

Yet Figure 2 suggest that with respect to N2O emission, only the lowest forest has significantly higher emissions. But the authors also imply in some places (including in the abstract) that there is a continuous gradient in N2O emissions. Is this in conflict with each other (Although probably having altitude as predictor may lead to statistically significant N2O gradients)?

Detailed (and minor) comments

Abstract L31: The statistical analysis does not show such a gradient, rather premontane forest was had much higher emissions than the rest (Figure 2). This may be a bit nit-picking on my part (I can see that the average in the lower montane forest is higher, but also has higher variability). Perhaps regress against altitude?)

Abstract L40: Is the sentence starting with "This bimodal.." is a bit empty, not add much

information. What is the complex relationship, what environmental variables?

Abstract L45: I think somewhere in the main text – perhaps discussion – it should be better laid out and evidenced that habitat is a proxy of NO3 availability.

L 95: check spelling "areally"

L 98: Sentence starting with "Nitrous oxide": the use of parenthesis seems odd.

L 104: Check the sentence – placement of "for" in the next line seems odd.

L 152: I like how the authors also analyzed topographic landforms. However, throughout the paper it is not clear, how these landforms were binned and weighted to form a habitat-wide data sets. Also, where were the samples taken from for the laboratory manipulations? Further, can the terminology be kept a bit more consistent? Throughout the manuscript, it is referred to as topography, landscape feature, landform, and basin landform. I assume they are all the same, but I suggest to use a consistent designation for this categorical variable.

L250: This sentence essentially repeats the statement in L240

L260: I assume the amount of litter added corresponds to the amount of litter falling in 1 month?

L483: Did you test for oxygen as a predictor, or was oxygen only assessed one time?

L506: >24 hour incubation: Over what period were the fluxes averaged?

L667: Again, how long is the >24h period?

L726: The figure shows that premontane habitat is significantly different from the other, and not that the lower elevation forests (premontane, and lower montane forest) are significantly different from the higher elevation forests.

L835: check the sentence starting with "Moreover,..."

L859: This sentence is not clear. What do the authors mean by "This pattern"

L884: It is hard to believe that NO3 additions did not stimulate N2O emission. Just eye-balling Fig 5 suggests, it seems that N2O flux over the incubation period increased with increasing NO3 levels added. Is there some artifact because of the way the ANOVA has been done (admittedly this is a weak point on my part – but maybe a recheck and some explanation is possible to enlighten me and the readers)?

Supplementary figure: Please add the habitat to the x-axis for completion

---

## Referee Comment (RC3) · Anonymous Referee #3 · 8 May 2017

Diem et al. report on a remarkably large and comprehensive set of observations and experiments examining N2O fluxes across the Kosnipata tropical elevation gradient in Peru. This was clearly a lot of work. The combination of high temporal resolution chamber observations with WFPS, 15N and litter experiments makes the study particularly compelling. I have four suggestions. First, there a few aspects of the 15N tracer work that require further clarification. Second, I recommend the authors consider scaling their observations to annual values. Third, depending on details of the 15N tracer methods, I suggest the authors consider making use of the N2: N2O flux ratios from the incubations to estimate total N gas losses from these ecosystems if appropriate. Finally, I think the authors could do a better job at contextualizing their work

with reference to other studies and its global implications.

15N tracers: It would appear that the WFPS experiment was not a true "tracer" experiment but is also a N addition experiment and is therefore confounded. For the lower elevation sites, 200 ug N/g soil is not trivial. Are you sure that the background NO3 values are correct? The reported NO3-N values from soil extractions of ∼150 ug/g are approximately 5-10 times higher than those observed in across most high N old-growth tropical forests worldwide. Tracer experiments often add < 0.5 ug/g at 15NO3 of ∼99 atom percent. Further, unless I missed it, there is no description of the isotopic enrichment levels (per mil or atom percent). This needs to be included.

Scaling: Given the seasonal representation of the sampling, I think annual scaling could be justified. When scaled annually, the mean N2O-N emissions (0.27 mg N m-2 day-1) would be ∼ 0.98 kg N ha-1 yr-1 with peak fluxes of ∼2.7 kg N ha-1 yr-1. On average, chamber studies and models find that N2O losses from undisturbed humid tropical soils are ∼1-4 kg N ha-1 yr-1 (See van Lent et al. Biogeosciences 2015 and Werner et al. Global Biogeochemical Cycles 2007). So, these values fit right in.

N2 fluxes: Given the response to the first point above, I suggest considering approximating total N gas losses from these ecosystems. Despite potential artifactual contributions of the incubations (disturbance, N additions) one could calculate rough N2 losses assuming equal N2:N2O ratios at a given WFPS as measured during the chamber work. This could be insightful as there are many chamber-based N2O estimates for tropical forests published but very few for total N gas fluxes because it's difficult to measure. Eyeballing the 15N2 versus 15N2O flux ratios (∼20 to 80) and applying these to the chamber observations would yield N2 fluxes of ∼20 − 216 kg N ha-1 yr-1. The lower-end flux is possible (see Fang et al. PNAS 2015) but the upper end estimate is highly unlikely. Such total N export rates could never persist in a near-equilibrium forest as even the lower end is higher than average N mineralization and annual plant uptake and far exceeds external N inputs in tropical forests (see Brookshire et al. Geophysical Research Letters 2017).

The beauty of the Kosnipata gradient is that it represents a quasi-space-for-climate change substitution. More could be done with this context in the introduction and discussion. Further there are many other papers examining denitrification in tropical landscapes (some of them mentioned here) that would benefit the narrative to include.

―――――――――――――――――――――

---

## Referee Comment (RC4) · Anonymous Referee #4 · 11 May 2017

Diem et al. present a comprehensive set of lab and field data relating to controls of soil nitrous oxide flux across an elevation gradient in the Peruvian Andes. As both long-term field measurements and lab-based manipulations are included, they are able to approach the discussion of N2O fluxes in these ecosystems from several different directions. This was excellent work that will be a valuable addition to our current knowledge of N-oxide fluxes and tropical montane ecosystems. However, the authors could really improve the paper by taking some additional time to craft a more integrated presentation/summation of their study. The results section, in particular, should be revised. A well-designed table or figure (or combination) could provide a fascinating and useful summation of the different experiments, while eliminating the repetitive text. Instead,

the text of the results section should highlight the most important results – much of this could be moved from the discussion section, which can then be condensed and re-focused to provide a bit more literature context about the different aspects of the results being discussed.

Minor comments: Line 105: substrates for _____? Line 138: give average temperature range over the course of the study Line 161: change 'because of' to 'due to' Line 172: provide volume of chamber Line 179: specify intervals Line 187-192: were zeroes included? Line 227-230: provide more detail: soil samples were taken in the field, air-dried and then re-wetted to target WFPS? Line 231-233: needs clarification: 0-10 cm depth included the organic layer at all elevations, except in the upper montane forest where 0-10 cm depth included only mineral? If 0-10 sometimes included the organic layer, what was the thickness of the organic layer at those elevations? What was the thickness of the organic layer at the upper montane site; how deep did you go to access the 0-10 mineral sample? explain reasoning behind this sampling decision; could this have affected your results? Line 297-307: clearly distinguish between 'soil core' and 'soil sample'; "core" implies that the soil is still intact – once it has been mixed and added to the jars, the soil samples are no longer soil cores Line 300-301: unclear; the five cores were mixed and then split into four equal parts? was the subsample and WFPS adjustment done on the cores or on the mixed soil in the jars? Line 375: change 'with' to 'and' Line 462: followed by topography Line 473: change 'is' to 'was' Line 474: define the fluctuation or refer to a table or figure where it is defined Line 585: change 'for' to 'from' Line 761: change semicolon to comma Line 768: between soil temperature and ___? Line 779: change 'as' to 'at' Line 782: change 'are' to 'is' Line 836: remove 'and'
* * *

---

## Editor Comment (EC1) · F. Joos (Editor) · 11 May 2017

Dear Authors

Your manuscript has been assessed by all four nominated referees. All referees find your study of interest and offer advice for further improvements of your manuscript.

I kindly ask you to prepare a reply to the referee's comments. The reply should also document how you will take into account the comments and criticisms in a revised manuscript.

The interactive discussion will stay open for the public until May 25 and additional comments may still be submitted by the scientific community.

[Figure]

Thank you for submitting your work to Biogeosciences.

With kind regards, Fortunat Joos
* * *

---

## Author Comment (AC1) · 15 Jun 2017

The authors would like to thank the referees and associate editor for the thoughtful and constructive remarks they have provided on our manuscript. We provide a detailed response to the referees' concerns in the sections below.

RESPONSE TO REFEREE 1

1. In 2014, some of the authors of the present publication published in Biogeosciences (doi:10.5194/bg-11-2325-2014) a paper entitled "Methane and nitrous oxide fluxes across an elevation gradient in the tropical Peruvian Andes". It was a very interesting

paper because there is only little information about soil nitrous oxide fluxes and their controls in tropical montane forest soils. In their one-year study they pointed out that nitrous oxide fluxes were primarily driven by denitrification and that nitrate availability was the principal constraint on soil nitrous oxide fluxes followed by soil moisture. In the present study Diem and colleagues extended their time-series to multi-annual time scales to identify controls of longer-term climatic variability, soil moisture and substrate availability on nitrous oxide fluxes in greater detail. They found out that habitat/elevation site, a proxy for nitrate availability under field conditions, was the best predictor for nitrous oxide fluxes. It is a great study. I have only few suggestions.

AUTHOR RESPONSE: Thank you to the referee for the positive remarks on our manuscript. We welcome the referees' suggestion for improvements to our manuscript.

2. I would suggest to reformulate the introduction and the hypotheses. The main message is that habitat/elevation – a proxy for NO3 availability in the field – is the best predictor for N2O flux and that seasonal differences of N2O flux and environmental variables were most pronounced at the lower montane forest site, where N2O flux was best explained by a combination of temperature, WFPS and N-availability. I would remove substrate availability and/or labile organic matter because it does not enrich the discussion but rather blur the main message. I think it is sufficient to discuss an absent correlation between N2O flux and variations in leaf-litter fall in one or two sentences and not in a whole discussion section (L827-L843).

AUTHOR RESPONSE: The referee makes a valuable observation about how the research is framed in the introduction, which is in-line with the suggestion of the second referee to reformulate our hypotheses so that they are more nuanced (point 17 below). With respect to the remark about removing the hypothesis about labile organic matter; we would prefer to keep this hypothesis, because several commonly used models use labile organic matter as a parameter for predicting N2O flux, including DAYCENT, DNDC, and the model ECOSSE (developed by co-author Smith) (Li, 2000;Smith et al., 2007;Werner et al., 2007). As a consequence, the negative finding from our field-based

litter manipulation is still an important result, because it suggests that labile organic matter may be a less important driver N2O flux in these montane tropical ecosystems. However, we are happy to streamline the discussion surrounding the litter manipulation study, in the interests of brevity.

3. At the moment it seems that results and discussion section are dominated by the description and interpretation of the experimental results in the lab. I am very sceptical whether the results from the laboratory-based nitrogen and WFPS manipulations can be directly linked to the results obtained in the field, especially when they are as puzzling and surprising as in the present study (i.e. WFPS-manipulation study). Substrate availability, nutrient limitations and a cascade of active microbial community composition may have drastically changed during transportation from the field site in Peru to Aberdeen. As long as there is no clearer picture about the active microbial community in the samples before and after transport, all of the nutrient and trace gas flux observations during incubation experiments have only potential implications. Additionally, the ratio of N2O to N2 production is pH-dependent. Did you check for potential pH changes upon transportation?

AUTHOR RESPONSE: The referee's point about potential treatment effects from handling, transportation, and storage of soils is well-made, and we recognise that the results from the laboratory experiments represent only the potential behaviour of the soils. As far as possible, we tried to minimise potential treatment effects by transporting soils under ambient (room temperature) conditions, recognising that cold storage of tropical soils has been found to significantly alter soil process rates (Arnold et al., 2008;Verchot, 1999). We also set-up the field experiments as quickly as possible after the soils were received in Aberdeen; normally within only one or two weeks after the soils' arrival. Lastly, the laboratory incubations were conducted with intact soils, rather than sieved soils or slurries, recognising that destruction of soil structure can alter biogeochemical process rates by changing redox gradients within aggregates and altering substrate competition among anaerobes (Sexstone et al., 1985;Teh and Silver, 2006).

However, it is important to note that we were dependent upon the laboratory studies to help interpret patterns detected in the field data (e.g. responses to changes in WFPS or N availability), because the complex controls on N2O production and emission made it difficult to establish clear empirical patterns between control variables (e.g. WFPS, inorganic N availability) and emissions. We will aim to clarify all these points in the revised version of the manuscript, and more clearly acknowledge the potential limitations of our study. With respect to the question of pH changes before and after transportation; we believe it is unlikely that transportation will have significantly altered pH because we compared data from soils measured in Peru (Zimmermann et al., 2012;Zimmermann et al., 2009a;Zimmermann et al., 2009b;Zimmermann et al., 2010) against samples that were measured after transportation in the UK, and average pH values did not appear to differ. For the lab experiments described here, we unfortunately did not measure pH measured after transportation, but only at the end of the experiments. The pH values measured at the end of the experiment were, on average, half a unit higher than the pH values measured for field soils.

4. What I find more fascinating is the observation of a negative relationship between WFPS and N2O flux in the field. The authors suggest that increasingly anaerobic conditions may stimulate N2O reductase activity and lead to greater denitrification to N2. This strengthens the assumption of Mueller et al. 2015 who suggested that gaseous N loss was likely dominated by N2 rather than N2O in Ecuadorian montane forest soils. Taken together, this finding may be generalized to tropical montane forest ecosystems.

AUTHOR RESPONSE: Thank you for the suggested reference; we will read the paper you have suggested, and aim to revise our discussion to include the insights gained from Mueller et al. (2015) so as to deepen the links to other parallel studies.

5. This leads me to another suggestion. Many parts of the discussion section read like a repetition or better description of the results section (e.g. L740-L760; L814-L818; L851-L858; L869-L876; L881-L891). Moreover, the links between different parts are laborious (e.g. L730-L734; L751-L755; L784-790; L880). I think it is necessary to

make the reading more "fluid". Many sentences in the results and discussion section begin with "For example" (e.g. L534, L620, L689, L745, L814). I think the discussion section would benefit if present results would be more interpreted in the light of recent publications (e.g. Baldos et al. 2015; Mueller et al. 2015; Nottingham et al. 2015).

AUTHOR RESPONSE: This point is well-taken, and is in-line with referee 2's suggestion that we should streamline the results section by only discussing the main findings and putting the other statistical results in tables (point 16 below). We will also read the references suggested, and revise the discussion to incorporate a wider discussion of recent literature.

6. L45-L48: This should also be mentioned in the conclusion section

AUTHOR RESPONSE: Editorial suggestion will be taken in the revised version of the text.

7. L98: ...derived from (missing word)

AUTHOR RESPONSE: Editorial suggestion will be taken in the revised version of the text.

8. L290: What is the sampling size of the background concentration measurements?

AUTHOR RESPONSE: We measured background concentrations once for every individual soil core, thus n=5 for each elevation

9. L300: What was the length of time between sampling and analysis?

AUTHOR RESPONSE: Samples were analysed no more than one week after the samples arrived in Aberdeen. Transport time from Peru to the UK varied between one and two weeks.

10. L827-L843: Remove heading and shorten section.

AUTHOR RESPONSE: Editorial suggestion will be taken in the revised version of the

text.

11. L880-L900: Does this section really enrich the discussion?

AUTHOR RESPONSE: The aim of this paragraph was to link the patterns in the field data with what we found in the laboratory experiments. We also speculated as to why the nitrate reducing microbes in our soils showed such a weak response to relatively large manipulations of inorganic N availability, given that we expected that the microbes would show a stronger short-term response to elevated N inputs.

12. L906-L907: "Nitrous oxide flux originated primarily from nitrate reduction rather than from nitrification, probably due to low pH soil condition". Influence of pH has not been discussed in previous sections.

AUTHOR RESPONSE: We will aim to introduce the concept of pH as an important controlling variable earlier in the text.

13. L912: It should be clearly stated whether results were obtained from incubation experiments or from the field. Table1, Figure 3: Table and figure are very difficult to read. May be you can upload tables and figures in a higher resolution.

AUTHOR RESPONSE: Editorial suggestion will be taken in the revised version of the text.

14. References: Baldos et al. 2015 (DOI: 10.1890/14-0295.1) Mueller et al. 2015 (DOI: 10.3389/feart.2015.00066) Nottingham et al. (DOI:10.5194/bg-12-6071-2015)

AUTHOR RESPONSE: These references will be incorporated in the revised version of the text, and their findings incorporated into our wider discussion of the results.

RESPONSE TO REFEREE 2

15. The authors address the complex issue of N2O emissions that is globally, even more for tropical forests, and particularly for montane tropical forests widely unconstrained. The experimental setup in the field and in the laboratory were designed to

capture mechanisms that affect N2O production and emissions. These effects include soil moisture, substrate availability (both mineral nitrogen and labile organic matter), soil moisture, oxygen, and temperature. They further analyzed more indirect predictors such as biome type, topography, seasonality, year to year variability as well as interacting effects among these potential drivers for N2O production. The major outcome of this study is that the controls on N2O emissions remain elusive and in parts counter existing knowledge. In particular, the study finds little seasonal variability despite strong seasonality in wetness. Further, soil moisture experiments suggest not the straightforward controls as they are being used in conceptual and numerical models. The exhaustive work done in soils in difficult and previously unsampled environment, as well as (in my view) important laboratory experiments that complement the field work. The data deserves dissemination to the scientific public. However, I do have some suggestions and comments on the presentation and interpretation of the data.

AUTHOR RESPONSE: Thank you to the referee for the positive remarks on our manuscript and the constructive comments provided below.

16. Organization: The sheer number of observations and experiments, the exhaustive statistical analysis makes, and the resulting (complex pattern) makes it hard to write a clean story. Yet I think the authors should give the presentation some more thought. The result section is full of statistical test results, I am wondering if the tests applied and their results would not be better confined to tables, while the result text focuses more on the most important patterns.

AUTHOR RESPONSE: Thank you for these useful suggestions. It was, admittedly, difficult to find a very simple and elegant way of presenting the data, given the diverse experiments and complex results. The referee's suggestion is well-taken, and in-line with the first referee's remarks (point 5 above). We will aim to revise the results so that the statistical outputs are summarised more neatly in tables, and only the major findings referenced in the main body of the text.

17. Hypotheses: I would love to see a bit more nuanced hypotheses: Teh et al., 2014 already show an "odd" relationship with soil moisture (i.e. unexpected highs during dry season compared to wet season). Could better hypotheses be developed based on this earlier data? In light of previous work done at the site, H1 and H2 are fairly generic. Similarly, since the paper also addresses elevation gradients (or transitions from premontane tropical forests to montane grasslands, perhaps there are potential to use that gradient to set up additional hypotheses (What are expectations if compared to [seasonally dry] lowland tropical systems?).

AUTHOR RESPONSE: This comment is also broadly in-line with observations made by referee 1 (point 2 above). We will revise the hypotheses in order to make them more nuanced, and to better-reference our earlier published study.

18. Seasonality: Looking at the time series, it seems to me from the get go there is no direct seasonal effect. However, there are curious seasonal patterns: Soil moisture seems to lag quite a bit the precipitation (i.e. soil moisture seems to increase at the beginning of the dry season before it diminishes, while soil moisture continues to decline after the onset of the wet season). Much harder to discern, but just eyeballing the data in Fig 3, it seems there is a seasonal pattern of N2O emissions that it out of phase with seasonality, and is also out of phase with soil moisture. I do not have a mechanistic explanation how such lags can be formed given that often the first rain leads to strong pulses in denitrification. Nor do I know whether the patterns I seem to recognize are really there if further scrutinized. Yet I am wondering if there should be some exploration with the inclusion of lag in the analysis. Perhaps the authors toyed with it and did not pan out, However, I would be curious to know either way.

AUTHOR RESPONSE: We analysed the data in a number of different ways in order to explore not only instantaneous but lagged responses of N2O flux to rainfall. Unfortunately, because we did not have large enough number of data points, we were unable to employ more sophisticated time series methods (e.g. AR, ARIMA) to evaluate whether or not the apparent lags in the data were real, and were reliant on more

simple methods of analysis such as repeated measures ANOVA. We were unable to pinpoint lag effects using this method of analysis, although this is not to say these lags do not in fact exist, merely that we were unable to detect them using the sampling method and analysis tools that we employed.

19. Bimodal soil moisture response: The authors put strong emphasis on the bimodal soil moisture response of N2O emissions with peaks at 90 % and 50 % water filled pore space – stating it both in the abstract and the conclusion. However, this is in my view not clearcut, occurring only in some of the sampled soils. The results and the discussion acknowledge this. Is there a way to nuance the abstract and conclusion, such that the result do not come over as overstated?

AUTHOR RESPONSE: We will aim revise the text to better highlight that this apparent bimodal trend is derived from the overall (pooled) dataset, and may not be universal or as straight forward for all study sites.

20. Gradient nitrogen-rich -> nitrogen poor. In several places there is mention that the premontane and the lower montane habitats are nitrogen rich, whereas the higher elevations are considered nitrogen poor. It is perhaps worthwile to define N rich and N poor explicitly (for example by resin bag mineral N). This seems to be very important, given that nitrate availability may be a strong driver for N2O production.

AUTHOR RESPONSE: The reference to N-rich or N-poor is relative, and largely in reference to the parallel Ecuadorian transect, where similar types of studies have been conducted by our German colleagues. We will revise the text to provide a better regional context for our statements.

21. Yet Figure 2 suggest that with respect to N2O emission, only the lowest forest has significantly higher emissions. But the authors also imply in some places (including in the abstract) that there is a continuous gradient in N2O emissions. Is this in conflict with each other (Although probably having altitude as predictor may lead to statistically significant N2O gradients)?

AUTHOR RESPONSE: We will revise the manuscript to clarify the point that there is not so much a gradient as a step change or shift in N2O flux from premontane forest to the other habitats.

22. Abstract L31: The statistical analysis does not show such a gradient, rather premontane forest was had much higher emissions than the rest (Figure 2). This may be a bit nit-picking on my part (I can see that the average in the lower montane forest is higher, but also has higher variability). Perhaps regress against altitude?)

AUTHOR RESPONSE: Editorial suggestion will be taken in the revised version of the text.

23. Abstract L40: Is the sentence starting with "This bimodal.." is a bit empty, not add much information. What is the complex relationship, what environmental variables?

AUTHOR RESPONSE: Editorial suggestion will be taken in the revised version of the text. We will revise the sentence structure to clarify meaning.

24. Abstract L45: I think somewhere in the main text – perhaps discussion – it should be better laid out and evidenced that habitat is a proxy of NO3 availability.

AUTHOR RESPONSE: Editorial suggestion will be taken in the revised version of the text.

25. L 95: check spelling "areally"

AUTHOR RESPONSE: Editorial suggestion will be taken in the revised version of the text.

26. L 98: Sentence starting with "Nitrous oxide": the use of parenthesis seems odd. L 104: Check the sentence – placement of "for" in the next line seems odd.

AUTHOR RESPONSE: Editorial suggestion will be taken in the revised version of the text. This is probably a typological/grammatical error that we missed when editing the manuscript.

27. L 152: I like how the authors also analyzed topographic landforms. However, through- out the paper it is not clear, how these landforms were binned and weighted to form a habitat-wide data sets. Also, where were the samples taken from for the laboratory manipulations? Further, can the terminology be kept a bit more consistent? Throughout the manuscript, it is referred to as topography, landscape feature, landform, and basin landform. I assume they are all the same, but I suggest to use a consistent designation for this categorical variable.

AUTHOR RESPONSE: Topography/landform was treated as a categorical variable in our repeated measures ANOVA or ANCOVA tests. For the laboratory incubations, two soil cores were sampled from each landform. With respect to terminology; we will go through the text a revise the language so that we keep to the same terminology so as to avoid reader confusion.

28. L250: This sentence essentially repeats the statement in L240

AUTHOR RESPONSE: This sentence will be removed in the revised version of the manuscript.

29. L260: I assume the amount of litter added corresponds to the amount of litter falling in 1 month?

AUTHOR RESPONSE: Yes.

30. L483: Did you test for oxygen as a predictor, or was oxygen only assessed one time?

AUTHOR RESPONSE: Soil oxygen content was measured every time soil gas flux was sampled.

31. L506: >24 hour incubation: Over what period were the fluxes averaged?

AUTHOR RESPONSE: The overall period for the incubation was 48 hours. For the late phase of the incubation, we calculated the flux rate over 24 to 48 hours.

32. L667: Again, how long is the >24h period?

AUTHOR RESPONSE: Please see point 31.

33. L726: The figure shows that premontane habitat is significantly different from the other, and not that the lower elevation forests (premontane, and lower montane forest) are significantly different from the higher elevation forests.

AUTHOR RESPONSE: The text will be revised to correct this oversight.

34. L835: check the sentence starting with "Moreover,. . ."

AUTHOR RESPONSE: Editorial suggestion will be taken in the revised version of the text.

35. L859: This sentence is not clear. What do the authors mean by "This pattern"

AUTHOR RESPONSE: We were referring to the overall trend of decreasing N2O flux with increasing elevation. We will revise this sentence to try and clarify our meaning.

36. L884: It is hard to believe that NO3 additions did not stimulate N2O emission. Just eye- balling Fig 5 suggests, it seems that N2O flux over the incubation period increased with increasing NO3 levels added. Is there some artifact because of the way the ANOVA has been done (admittedly this is a weak point on my part – but maybe a recheck and some explanation is possible to enlighten me and the readers)?

AUTHOR RESPONSE: When evaluating for the effect of N addition level on N2O flux, the ANOVA pooled data across all other categories (i.e. site, incubation phase) to compare the difference in N2O flux among N treatments. Because of the high level of variability in N2O flux among study sites and incubation phases, the net effect was that the ANOVA found no clear signal of N addition level alone. The lack of trend is not an artefact of the ANOVA calculation per se, but rather represents the high level of variability among soils from different study sites and differing responses of N2O flux during different incubation phases.
37. Supplementary figure: Please add the habitat to the x-axis for completion

AUTHOR RESPONSE: Editorial suggestion will be taken in the revised version of the text.

RESPONSE TO REFEREE 3

38. Diem et al. report on a remarkably large and comprehensive set of observations and experiments examining N2O fluxes across the Kosnipata tropical elevation gradient in Peru. This was clearly a lot of work. The combination of high temporal resolution chamber observations with WFPS, 15N and litter experiments makes the study particularly compelling. I have four suggestions. First, there a few aspects of the 15N tracer work that require further clarification. Second, I recommend the authors consider scaling their observations to annual values. Third, depending on details of the 15N tracer methods, I suggest the authors consider making use of the N2: N2O flux ratios from the incubations to estimate total N gas losses from these ecosystems if appropriate. Finally, I think the authors could do a better job at contextualizing their work with reference to other studies and its global implications.

AUTHOR RESPONSE: We thank the referee for the positive remarks on our manuscript. We will endeavour, in the revised version of the manuscript, to address the four key suggestions that the referee has outlined here, as we believe that the referee's suggestions will enable us to produce a manuscript that better frames and contextualises our research in a wider regional and global context.

39. 15N tracers: It would appear that the WFPS experiment was not a true "tracer" experiment but is also a N addition experiment and is therefore confounded. For the lower elevation sites, 200 ug N/g soil is not trivial. Are you sure that the background NO3 values are correct? The reported NO3-N values from soil extractions of âĹij150 ug/g are approximately 5-10 times higher than those observed in across most high N old- growth tropical forests worldwide. Tracer experiments often add < 0.5 ug/g at 15NO3 of âĹij99 atom percent. Further, unless I missed it, there is no description of

the isotopic enrichment levels (per mil or atom percent). This needs to be included.

AUTHOR RESPONSE: Upon closer inspection, we realised that the values reported in the table are incorrect, and that the actual amount of N added was in fact much smaller than reported in the text. For example, for the WFPS experiment, the added amounts were 200 ng N/g soil for the lower elevation sites and 20 ng N/g soil for the higher elevation ones. For the N addition experiment, the values of N reported are the total amount of N added for the soil sample, and need to be normalised so that the values are reported on a per g soil basis. Thus, the true amounts of 15N tracer added in both the WFPS and N addition experiments are in fact in-line with the "trace" amount more typical of these types of 15N labelling experiments. We will correct this in the revised version of the text. With respect to the level of isotopic enrichment; we applied the tracers at a 30 atom % level (see line 309). We will add this information to the description of the WFPS experiment to ensure clarity of expression.

40. Scaling: Given the seasonal representation of the sampling, I think annual scaling could be justified. When scaled annually, the mean N2O-N emissions (0.27 mg N m-2 day-1) would be âĹij 0.98 kg N ha-1 yr-1 with peak fluxes of âĹij2.7 kg N ha-1 yr-1. On average, chamber studies and models find that N2O losses from undisturbed humid tropical soils are âĹij1-4 kg N ha-1 yr-1 (See van Lent et al. Biogeosciences 2015 and Werner et al. Global Biogeochemical Cycles 2007). So, these values fit right in.

AUTHOR RESPONSE: In our earlier paper (Teh et al., 2014), we provided area-weighted annual estimates, and are happy to conduct a similar simple extrapolation exercise here to provide area-weighted seasonal flux estimates. While these simple estimates are not as robust as those derived from process-based models, they are useful in terms of furthering the discussion about the potential role that these types of montane tropical ecosystems play in regional and global atmospheric budgets.

41. N2 fluxes: Given the response to the first point above, I suggest considering approximating total N gas losses from these ecosystems. Despite potential artifactual

contributions of the incubations (disturbance, N additions) one could calculate rough N2 losses assuming equal N2:N2O ratios at a given WFPS as measured during the chamber work. This could be insightful as there are many chamber-based N2O estimates for tropical forests published but very few for total N gas fluxes because it's difficult to measure. Eyeballing the 15N2 versus 15N2O flux ratios (âĹij20 to 80) and applying these to the chamber observations would yield N2 fluxes of âĹij20 – 216 kg N ha-1 yr-1. The lower-end flux is possible (see Fang et al. PNAS 2015) but the upper end estimate is highly unlikely. Such total N export rates could never persist in a near-equilibrium forest as even the lower end is higher than average N mineralization and annual plant uptake and far exceeds external N inputs in tropical forests (see Brookshire et al. Geophysical Research Letters 2017).

AUTHOR RESPONSE: We will take this into consideration, and will calculate potential N2 emissions for our sites from the N2O/N2 ratios. However, as the referee already suggests, these values, at the upper end of the range, may be unreasonably high, most likely due to the fact that the conditions for the laboratory experiments do not match conditions in situ, and may overestimate field flux rates. However, this exercise may be useful in generating wider discussion in the community, and enabling us to develop hypotheses for future testing.

42. The beauty of the Kosnipata gradient is that it represents a quasi-space-for-climate change substitution. More could be done with this context in the introduction and discussion. Further there are many other papers examining denitrification in tropical landscapes (some of them mentioned here) that would benefit the narrative to include.

AUTHOR RESPONSE: This remark is in agreement with concerns raised by other referees, and we will improve the introduction and discussion to include more recent research on this topic.

RESPONSE TO REFEREE 4

43. Diem et al. present a comprehensive set of lab and field data relating to controls

of soil nitrous oxide flux across an elevation gradient in the Peruvian Andes. As both long-term field measurements and lab-based manipulations are included, they are able to approach the discussion of N2O fluxes in these ecosystems from several different directions. This was excellent work that will be a valuable addition to our current knowledge of N-oxide fluxes and tropical montane ecosystems. However, the authors could really improve the paper by taking some additional time to craft a more integrated presentation/summation of their study. The results section, in particular, should be revised. A well-designed table or figure (or combination) could provide a fascinating and useful summation of the different experiments, while eliminating the repetitive text. Instead, the text of the results section should highlight the most important results – much of this could be moved from the discussion section, which can then be condensed and re-focused to provide a bit more literature context about the different aspects of the results being discussed.

AUTHOR RESPONSE: Thank you for these very positive remarks. The referee's suggestions for improvement are broadly in-line with other referees' concerns (e.g. point 3) about the manner in which the results have been presented, and we will revise the text to address these concerns.

44. Line 105: substrates for ____?

AUTHOR RESPONSE: The sentence should have been deleted while editing; the revised text will be changed accordingly.

45. Line 138: give average temperature range over the course of the study

AUTHOR RESPONSE: Mean annual temperature is provided in Table 1.

46. Line 161: change 'because of' to 'due to'

AUTHOR RESPONSE: Editorial suggestion will be taken in the revised version of the text.

47. Line 172: provide volume of chamber

AUTHOR RESPONSE: The chamber volume was approximately 0.008m3 (8 L); the text will be revised accordingly.

48. Line 179: specify intervals

AUTHOR RESPONSE: Gas samples were collected at evenly spaced intervals over a 30 minute period; i.e. samples were collected 7.5 minutes apart. The text will be revised accordingly.

49. Line 187-192: were zeroes included?

AUTHOR RESPONSE: Yes. The revised text will be altered accordingly.

50. Line 227-230: provide more detail: soil samples were taken in the field, air-dried and then re-wetted to target WFPS?

AUTHOR RESPONSE: To clarify, the WFPS experiments were conducted with field-moist samples; i.e. the soil samples were collected from the field, shipped to Aberdeen, and subsequently distributed into glass jars without being fully air-dried. For incubations where the target WFPS was below the field moisture levels, the soils were allowed to partially air-dry until they reached a value 10 % below the target WFPS for the experiment, and then carefully re-wetted through the 15N tracer application to bring up the soil moisture up to the target levels. For treatments where the target WFPS was above field moisture levels, the soils were simply wetted to 10 % below the target WFPS and then the 15N tracer solution added to bring the soil up to the target moisture level. The text will be revised to describe this procedure in greater depth.

51. Line 231-233: needs clarification: 0-10 cm depth included the organic layer at all elevations, except in the upper montane forest where 0-10 cm depth included only mineral? If 0-10 sometimes included the organic layer, what was the thickness of the organic layer at those elevations? What was the thickness of the organic layer at the upper montane site; how deep did you go to access the 0-10 mineral sample? explain reasoning behind this sampling decision; could this have affected your results?

AUTHOR RESPONSE: For premontane forest, lower montane forest, and montane grassland, the organic matter in the upper 10 cm soil layer is intermixed with the mineral phase, and does not constitute a distinct mineral-free horizon. Thus, we simply sampled from the 0-10 cm depth because there was no practical means of separating the organic matter from the mineral soil in these habitats. In contrast, upper montane forest soil shows a very different pattern of vertical stratification compared to the other habitats. In this habitat, the mineral soil is overlain by a thick (up to 17 cm deep) mineral-free organic layer, consisting of poorly decomposed leaves, roots, and humic materials; very akin to low density peat. To sample the mineral soil in this habitat, we went below this distinct organic horizon to a depth of approximately 17 cm.

With respect to the WFPS experiment; we decided to collect mineral soil from below the organic horizon in the upper montane forest because there was no mineral material found in this layer, making it difficult to compare results between habitats (given that the other habitats contain mineral material in the upper 10 cm of their soil profiles). At the time, we did not consider sampling the organic layer as well. This was an oversight on our part, which we tried to partially correct in our N addition experiments, by including the organic layer in those subsequent experiments. In the revised form of the manuscript, we will try to acknowledge this shortcoming in our laboratory experiments.

52. Line 297-307: clearly distinguish between 'soil core' and 'soil sample'; "core" implies that the soil is still intact – once it has been mixed and added to the jars, the soil samples are no longer soil cores

AUTHOR RESPONSE: Editorial suggestion will be taken in the revised version of the text.

53. Line 300-301: unclear; the five cores were mixed and then split into four equal parts? was the subsample and WFPS adjustment done on the cores or on the mixed soil in the jars?

AUTHOR RESPONSE: Each of the cores was split into four equal parts. The text will be re-written to clarify this point.

54. Line 375: change 'with' to 'and'

AUTHOR RESPONSE: Editorial suggestion will be taken in the revised version of the text.

55. Line 462: followed by topography

AUTHOR RESPONSE: Editorial suggestion will be taken in the revised version of the text.

56. Line 473: change 'is' to 'was'

AUTHOR RESPONSE: Editorial suggestion will be taken in the revised version of the text.

57. Line 474: define the fluctuation or refer to a table or figure where it is defined

AUTHOR RESPONSE: Editorial suggestion will be taken in the revised version of the text.

58. Line 585: change 'for' to 'from'

AUTHOR RESPONSE: Editorial suggestion will be taken in the revised version of the text.

59. Line 761: change semicolon to comma

AUTHOR RESPONSE: Editorial suggestion will be taken in the revised version of the text.

60. Line 768: between soil temperature and ___?

AUTHOR RESPONSE: N2O

61. Line 779: change 'as' to 'at'

AUTHOR RESPONSE: Editorial suggestion will be taken in the revised version of the text.

62. Line 782: change 'are' to 'is'

AUTHOR RESPONSE: Editorial suggestion will be taken in the revised version of the text.

63. Line 836: remove 'and'

AUTHOR RESPONSE: Editorial suggestion will be taken in the revised version of the text.

REFERENCES

Arnold, J., Corre, M. D., and Veldkamp, E.: Cold storage and laboratory incubation of intact soil cores do not reflect in-situ nitrogen cycling rates of tropical forest soils, Soil Biol. Biochem., 40, 2480-2483, 10.1016/j.soilbio.2008.06.001, 2008. Li, C. S.: Modeling trace gas emissions from agricultural ecosystems. , Nutrient Cycling in Agroecosystems, 58, 259-276, 2000. Sexstone, A. J., Revsbech, N. P., Parkin, T. B., and Tiedge, J. M.: Direct Measurement of Oxygen Profiles and Denitrification Rates in Soil Aggregates., Soil Sci. Soc. Am. J., 49, 645-651, 1985. Smith, P., Smith, J. U., Flynn, H., Killham, K., Rangel-Castro, I., Foereid, B., Aitkenhead, M., Chapman, S., Towers, W., Bell, J., Lumsdon, D., Milne, R., Thomson, A., Simmons, I., Skiba, U., Reynolds, B., Evans, C., Frogbrook, Z., Bradley, I., Whitmore, A., and Falloon, P.: ECOSSE: Estimating Carbon in Organic Soils - Sequestration and Emissions. Final Report., Scottish Executive Environment and Rural Affairs Department Report, 166 pp., 2007. Teh, Y. A., and Silver, W. L.: Effects of soil structure destruction on methane production and carbon partitioning between methanogenic pathways in tropical rain forest soils, Journal of Geophysical Research: Biogeosciences, 111, n/a-n/a, 10.1029/2005JG000020, 2006. Verchot, L. V.: Cold storage of a tropical soil decreases nitrification potential, Soil Sci. Soc. Am. J., 63, 1942-1944, 1999. Werner, C., Butterbach-Bahl, K., Haas,

E., Hickler, T., and Kiese, R.: A global inventory of N2O emissions from tropical rainforest soils using a detailed biogeochemical model, Global Biogeochemical Cycles, 21, 1-18, Gb3010 10.1029/2006gb002909, 2007. Teh, Y. A., Diem, T., Jones, S., Huaraca Quispe, L. P., Baggs, E., Morley, N., Richards, M., Smith, P., and Meir, P.: Methane and nitrous oxide fluxes across an elevation gradient in the tropical Peruvian Andes, Biogeosciences, 11, 2325-2339, 10.5194/bg-11-2325-2014, 2014. Zimmermann, M., Meir, P., Bird, M., Malhi, Y., and Ccahuana, A.: Litter contribution to diurnal and annual soil respiration in a tropical montane cloud forest, Soil Biology and Biochemistry, 41, 1338-1340, 2009a. Zimmermann, M., Meir, P., Bird, M. I., Malhi, Y., and Ccahuana, A. J. Q.: Climate dependence of heterotrophic soil respiration from a soil-translocation experiment along a 3000 m tropical forest altitudinal gradient, European Journal of Soil Science, 60, 895-906, 10.1111/j.1365-2389.2009.01175.x, 2009b. Zimmermann, M., Meir, P., Bird, M. I., Malhi, Y., and Ccahuana, A. J. Q.: Temporal variation and climate dependence of soil respiration and its components along a 3000 m altitudinal tropical forest gradient, Global Biogeochemical Cycles, 24, GB4012, 4011-4013, 10.1029/2010GB003787, 2010. Zimmermann, M., Leifeld, J., Conen, F., Bird, M. I., and Meir, P.: Can composition and physical protection of soil organic matter explain soil respiration temperature sensitivity?, Biogeochemistry, 107, 423-436, 10.1007/s10533-010-9562-y, 2012.

---

## Author Response (AR1)

Dear Associate Editor and Referees

On behalf of my co-authors, I would like to thank you for your thoughtful and constructive remarks, and for providing us with the opportunity to improve our manuscript. The text has now been thoroughly revised in order to meet the referees' concerns, with detailed responses to specific referee comments provided in the sections below. The Introduction and hypotheses have been re-written in order to draw stronger links between our prior research and the work presented here. We have also reformulated the hypotheses so that they more clearly test key ideas that emerged from our prior work. The Results have been simplified, particularly the sections pertaining to the laboratory manipulations, in order to aid reader understanding. Lastly, we have completely overhauled the Discussion, in order to better-integrate the discussion about the findings of our field and laboratory experiments. In addition, we have added a section discussing the implications of our measurements for the annual flux of N2O and N2, in-line with the recommendations of one of the referees. Please note that all line numbers referred to in this document are taken from the "clean" version of the text, where the "track changes" function in Microsoft Word has been disabled.

We hope that these changes will meet with your approval, and look forward to hearing from you in due course.

Yours sincerely,

Yit Arn Teh

RESPONSE TO REFEREE 1

1.      In 2014, some of the authors of the present publication published in Biogeosciences (doi:10.5194/bg-11-2325-2014) a paper entitled "Methane and nitrous oxide fluxes across an elevation gradient in the tropical Peruvian Andes". It was a very interesting paper because there is only little information about soil nitrous oxide fluxes and their controls in tropical montane forest soils. In their one-year study they pointed out that nitrous oxide fluxes were primarily driven by denitrification and that nitrate availability was the principal constraint on soil nitrous oxide fluxes followed by soil moisture. In the present study Diem and colleagues extended their time-series to multi-annual time scales to identify controls of longer-term climatic variability, soil moisture and substrate availability on nitrous oxide fluxes in greater detail. They found out that habitat/elevation site, a proxy for nitrate availability under field conditions, was the best predictor for nitrous oxide fluxes. It is a great study. I have only few suggestions.

AUTHOR RESPONSE: We thank the referee for the positive remarks on our manuscript and constructive suggestions provided below.

2.      I would suggest to reformulate the introduction and the hypotheses. The main message is that habitat/elevation – a proxy for NO3 availability in the field – is the best predictor for N2O flux and that seasonal differences of N2O flux and environmental variables were most pronounced at the lower montane forest site, where N2O flux was best explained by a combination of temperature, WFPS and N-availability. I would remove substrate availability and/or labile organic matter because it does not enrich the discussion but rather blur the main message. I think it is sufficient to discuss an absent correlation between N2O flux and variations in leaf-litter fall in one or two sentences and not in a whole discussion section (L827-L843).

AUTHOR RESPONSE: The referee makes a valuable observation about how the research is framed in the introduction, which is in-line with the suggestions of the second referee (point 17 below). The Introduction has now been heavily revised, in order to more clearly outline the key knowledge gaps identified by our earlier work, and to better establish the links between identified knowledge gaps and the research described in this manuscript (lines 99-153). The hypotheses have also been reformulated so that they link more explicitly to the unknowns and unresolved questions raised by our prior study, following the recommendations of the second referee (lines 134-142; also see point 17 below).

The only recommendation we have not fully acted upon is the suggestion the referee made with respect to the hypothesis on substrate limitation and labile organic matter (H3). With all due respect to the referee, we chose to retain this part of the hypothesis, because of the important role that labile organic matter is thought to play in modulating nitrate reduction (Morley and Baggs, 2010;Blackmer and Bremner, 1978;Davidson, 1991;Firestone et al., 1980;Weier et al., 1993). Moreover, the availability of labile organic matter is often used as a key input parameter for predicting N2O flux in several commonly used process-based models, such as DAYCENT, DNDC, and ECOSSE (Li, 2000;Smith et al., 2007;Werner et al., 2007). As a consequence, we believe that the negative finding from our field-based litter manipulation is still an important result to report on, because it suggests that labile organic matter may be a less important driver of N2O flux in these montane tropical ecosystems. However, we have acted on the referee's suggestion to condense the discussion of about labile organic matter so as not to belabour the point (lines 787-792).

3.      At the moment it seems that results and discussion section are dominated by the description and interpretation of the experimental results in the lab. I am very sceptical whether the results from the laboratory-based nitrogen and WFPS manipulations can be directly linked to the results obtained in the field, especially when they are as puzzling and surprising as in the present study (i.e. WFPS-manipulation study). Substrate availability, nutrient limitations and a cascade of active microbial community composition may have drastically changed during transportation from the field site in Peru to Aberdeen. As long as there is no clearer picture about the active microbial community in the samples before and after transport, all of the nutrient and trace gas flux observations during incubation experiments have only potential implications. Additionally, the ratio of N2O to N2 production is pH-dependent. Did you check for potential pH changes upon transportation?

AUTHOR RESPONSE: We recognise that the results from the laboratory experiments represent only the potential behaviour of these soils. However, the laboratory experiments were an important aid to understanding patterns in the field data because it was difficult to establish clear empirical relationships between control variables and N2O flux, due to the confounding effects of multiple environmental controls. This point has now been clarified in the revised version of the text (lines 123-129). Furthermore, we have revised the Discussion so that the discussion of the field and laboratory results are better integrated, to provide a more holistic view of how environmental factors control N2O flux (please see the newly revised sections 6.1 and 6.2). By integrating the discussion of field and laboratory results, we hope that presentation of the findings does not appear so heavily dominated by our laboratory experiments.

The referee's point about potential treatment effects from handling, transportation, and storage of soils is well made. As far as possible, we tried to minimize potential treatment effects by transporting soils under ambient (room temperature) conditions, recognising that cold storage of tropical soils has been found to significantly alter soil process rates (Arnold et al., 2008;Verchot, 1999). We also set-up the laboratory experiments as quickly as possible after the soils were received in Aberdeen, normally within one or two weeks after the soils' arrival. Lastly, the laboratory incubations were conducted with intact soils, rather than sieved soils or slurries, recognizing that destruction of soil structure can alter biogeochemical process rates by changing redox gradients within aggregates and altering substrate competition among anaerobes (Sexstone et al., 1985;Teh and Silver, 2006).

With respect to the question of pH changes before and after transportation; we believe it is unlikely that transportation will have significantly altered pH, because average pH values did not appear to differ when we compared data from soils measured in Peru (Zimmermann et al., 2012;Zimmermann et al., 2009a;Zimmermann et al., 2009b) against samples that were measured after transportation to the UK. For the lab experiments described here, we did not measure pH measured after transportation, but only at the end of the incubations. The pH values measured at the end of the incubations were, on average, half a unit higher than the pH values measured for field soils.

4.      What I find more fascinating is the observation of a negative relationship between WFPS and N2O flux in the field. The authors suggest that increasingly anaerobic conditions may stimulate N2O reductase activity and lead to greater denitrification to N2. This strengthens the assumption of Mueller et al. 2015 who suggested that gaseous N loss was likely dominated by N2 rather than N2O in Ecuadorian montane forest soils. Taken together, this finding may be generalized to tropical montane forest ecosystems.

AUTHOR RESPONSE: Thank you for the suggested reference; this paper and the insights gained from it have now been incorporated into the revised version of the text. The Discussion section was heavily revised to incorporate some of the more recent publications in this topic area, and efforts have been made to stress the wider pan-Andean patterns which may be emerging from studies in both Peru and Ecuador (please see the newly revised sections 6.1 to 6.3).

5.      This leads me to another suggestion. Many parts of the discussion section read like a repetition or better description of the results section (e.g. L740-L760; L814-L818; L851-L858; L869-L876; L881-L891). Moreover, the links between different parts are laborious (e.g. L730-L734; L751-L755; L784-790; L880). I think it is necessary to make the reading more "fluid". Many sentences in the results and discussion section begin with "For example" (e.g. L534, L620, L689, L745, L814). I think the discussion section would benefit if present results would be more interpreted in the light of recent publications (e.g. Baldos et al. 2015; Mueller et al. 2015; Nottingham et al. 2015).

AUTHOR RESPONSE: This point is well-taken, and is in-line with referee 2's suggestion that we should also streamline the results section (please see point 16 below). As noted in point 4 above, the Discussion has been completely overhauled in order to clarify some of the main messages, highlight commonalities between this study and parallel experiments elsewhere in the Andes, and in order to avoid undue repetition of information from the Results (please see the newly revised sections 6.1 to 6.3).

6.      L45-L48: This should also be mentioned in the conclusion section

AUTHOR RESPONSE: Editorial suggestion taken (lines 895-897).

7.      L98: ...derived from (missing word)

AUTHOR RESPONSE: The phrase "nitrate reduction" has now been added to the revised manuscript (line 105).

8.      L290: What is the sampling size of the background concentration measurements?

AUTHOR RESPONSE: We measured background concentrations once for every individual soil core, thus n=5 for each elevation. The text has now been revised to incorporate this information (line 329).

9.      L300: What was the length of time between sampling and analysis?

AUTHOR RESPONSE: Samples were analysed no more than one week after the samples arrived in Aberdeen. Transport time from Peru to the UK varied between one and two weeks. This information has now been added to the revised version of the text (lines 339-340).

10.     L827-L843: Remove heading and shorten section.

AUTHOR RESPONSE: Editorial suggestion taken; also see point 2 above (lines 787-792).

11.     L880-L900: Does this section really enrich the discussion?

AUTHOR RESPONSE: We believe so, because the aim of this paragraph was to link the patterns in the field data with what we found in the laboratory experiments. We also speculated as to why the nitrate reducing microbes in our soils showed such a weak response to relatively large manipulations of inorganic N availability, given that we expected that the microbes would show a stronger short-term response to elevated N inputs.

12.     L906-L907: "Nitrous oxide flux originated primarily from nitrate reduction rather than from nitrification, probably due to low pH soil condition". Influence of pH has not been discussed in previous sections.

AUTHOR RESPONSE: The Discussion has now been revised to include a discussion of how pH may influence N2O production from ammonia-oxidation; namely, that under acidic conditions, recent advances in soil microbial research indicate that ammonia oxidation is primarily driven by ammonia-oxidizing archaea, which produces relatively little N2O compared to ammonia-oxidizing bacteria (AOB) (Hink et al., 2016;Prosser and Nicol, 2008). As a consequence, under the acidic soil conditions observed here, we believe suspect that most of the N2O is derived from nitrate reduction since N2O production from nitrification is so meagre (lines 731-758).

13.     L912: It should be clearly stated whether results were obtained from incubation experiments or from the field.

AUTHOR RESPONSE: We have attempted to re-phrase the Conclusion so that it is clearer that these inferences are drawn from field observations (section 7).

14.     Table1, Figure 3: Table and figure are very difficult to read. May be you can upload tables and figures in a higher resolution.

AUTHOR RESPONSE: Table 1 and Figure 3 are now presented as higher resolution images in the revised text.

15.     References: Baldos et al. 2015 (DOI: 10.1890/14-0295.1) Mueller et al. 2015 (DOI: 10.3389/feart.2015.00066) Nottingham et al. (DOI:10.5194/bg-12-6071-2015)

AUTHOR RESPONSE: These references have been incorporated into the revised version of the text (see sections 6.1 to 6.3)

RESPONSE TO REFEREE 2

16.     The authors address the complex issue of N2O emissions that is globally, even more for tropical forests, and particularly for montane tropical forests widely unconstrained. The experimental setup in the field and in the laboratory were designed to capture mechanisms that affect N2O production and emissions. These effects include soil moisture, substrate availability (both mineral nitrogen and labile organic matter), soil moisture, oxygen, and temperature. They further analyzed more indirect predictors such as biome type, topography, seasonality, year to year variability as well as interacting effects among these potential drivers for N2O production. The major outcome of this study is that the controls on N2O emissions remain elusive and in parts counter existing knowledge. In particular, the study finds little seasonal variability despite strong seasonality in wetness. Further, soil moisture experiments suggest not the straightforward controls as they are being used in conceptual and numerical models. The exhaustive work done in soils in difficult and previously unsampled environment, as well as (in my view) important laboratory experiments that complement the field work. The data deserves dissemination to the scientific public. However, I do have some suggestions and comments on the presentation and interpretation of the data.

AUTHOR RESPONSE: We thank the referee for the positive remarks on our manuscript and constructive suggestions provided below.

17.    Organization: The sheer number of observations and experiments, the exhaustive statistical analysis makes, and the resulting (complex pattern) makes it hard to write a clean story. Yet I think the authors should give the presentation some more thought. The result section is full of statistical test results, I am wondering if the tests applied and their results would not be better confined to tables, while the result text focuses more on the most important patterns.

AUTHOR RESPONSE: Thank you for these useful suggestions. It was, admittedly, difficult to find a very simple and elegant way of presenting the data, given the large number of observations, manipulative experiments, and complex results. The referee's suggestion, however, is well-taken, and is in agreement with the first referee's remarks about simplifying the text and clarifying the message (please see point 5 above). To address the referee's concerns, the Results section of the text has now been extensively revised and shortened, so that only the most important findings of our research are presented in the main body of the text. We have concentrated our efforts on revising the sections of the text that pertain to the laboratory incubations (sections 5.4 and 5.6), because these experiments show the most complex experimental design (i.e. three-way full factorial ANOVA). Statistical outputs for these laboratory experiment have now been summarised in two new tables for ease of reference (Supplementary Online Materials Tables S2, and S3). For the field data (sections 5.1 and 5.2), we have also made subtle alterations to the sentence structure, and judiciously removed unnecessary text. We have also produced new tables summarising the outputs from our statistical analyses in order to facilitate clarity of understanding (Supplementary Online Materials Tables S1).

18.    Hypotheses: I would love to see a bit more nuanced hypotheses: Teh et al., 2014 already show an "odd" relationship with soil moisture (i.e. unexpected highs during dry season compared to wet season). Could better hypotheses be developed based on this earlier data? In light of previous work done at the site, H1 and H2 are fairly generic. Similarly, since the paper also addresses elevation gradients (or transitions from premontane tropical forests to montane grasslands, perhaps there are potential to use that gradient to set up additional hypotheses (What are expectations if compared to [seasonally dry] lowland tropical systems?).

AUTHOR RESPONSE: Thank you for this remark. This comment is broadly in-line with observations made by referee 1 (please see point 2 above). As discussed previously, the introduction and hypotheses have now been heavily revised to draw stronger and more explicit links between our prior work and the findings of this study. The hypotheses themselves have been reformulated to better-reference the knowledge gaps and unknowns identified in our prior research.

19. Seasonality: Looking at the time series, it seems to me from the get go there is no direct seasonal effect. However, there are curious seasonal patterns: Soil moisture seems to lag quite a bit the precipitation (i.e. soil moisture seems to increase at the beginning of the dry season before it diminishes, while soil moisture continues to de- cline after the onset of the wet season). Much harder to discern, but just eyeballing the data in Fig 3, it seems there is a seasonal pattern of N2O emissions that it out of phase with seasonality, and is also out of phase with soil moisture. I do not have a mechanistic explanation how such lags can be formed given that often the first rain leads to strong pulses in denitrification. Nor do I know whether the patterns I seem to recognize are really there if further scrutinized. Yet I am wondering if there should be some exploration with the inclusion of lag in the analysis. Perhaps the authors toyed with it and did not pan out, However, I would be curious to know either way.

AUTHOR RESPONSE: We analysed the data in a number of different ways in order to explore not only instantaneous but lagged responses of N2O flux to rainfall. Unfortunately, because we did not have large enough number of data points, we were unable to employ more sophisticated time series approaches, such as autoregressive models, to evaluate whether the apparent lags in the data were real. We were therefore reliant on more simple methods of analysis, such as repeated measures ANOVA. We were unable to pinpoint lag effects using this method of analysis, although this is not to say these lags do not in fact exist; merely that we were unable to detect them using the sampling method and analysis tools that we employed.

20. Bimodal soil moisture response: The authors put strong emphasis on the bimodal soil moisture response of N2O emissions with peaks at 90 % and 50 % water filled pore space – stating it both in the abstract and the conclusion. However, this is in my view not clearcut, occurring only in some of the sampled soils. The results and the discussion acknowledge this. Is there a way to nuance the abstract and conclusion, such that the result do not come over as overstated?

AUTHOR RESPONSE: The manuscript has now been revised to clarify that this general bimodal trend is apparent only in the pooled dataset, subtly implying that there may be uncertainty as to whether this general trend is applicable for individual habitats (line 44).

21.    Gradient nitrogen-rich -> nitrogen poor. In several places there is mention that the premontane and the lower montane habitats are nitrogen rich, whereas the higher elevations are considered nitrogen poor. It is perhaps worthwile to define N rich and N poor explicitly (for example by resin bag mineral N). This seems to be very important, given that nitrate availability may be a strong driver for N2O production.

AUTHOR RESPONSE: Thank you for this suggestion. The Discussion has now been revised to reference the resin-extractable nitrate data in order to better anchor the comparisons against a more objective empirical index (section 6.2).

22.    Yet Figure 2 suggest that with respect to N2O emission, only the lowest forest has significantly higher emissions. But the authors also imply in some places (including in the abstract) that there is a continuous gradient in N2O emissions. Is this in conflict with each other (Although probably having altitude as predictor may lead to statistically significant N2O gradients)?

AUTHOR RESPONSE: We apologize for this error. The manuscript has now been revised to improve the precision of our language (e.g. lines 33-34).

23.    Abstract L31: The statistical analysis does not show such a gradient, rather premontane forest was had much higher emissions than the rest (Figure 2). This may be a bit nit-picking on my part (I can see that the average in the lower montane forest is higher, but also has higher variability). Perhaps regress against altitude?)

AUTHOR RESPONSE: Editorial suggestion taken; please see point 22.

24.    Abstract L40: Is the sentence starting with "This bimodal.." is a bit empty, not add much information. What is the complex relationship, what environmental variables?

AUTHOR RESPONSE: Editorial suggestion taken; the phrases "bimodal distribution" and "environmental variables" have been removed (lines 46-47).

25.    Abstract L45: I think somewhere in the main text – perhaps discussion – it should be better laid out and evidenced that habitat is a proxy of NO3 availability.

AUTHOR RESPONSE: The case that habitat is a proxy for NO3- availability is now made in lines 794-815.

26.    L 95: check spelling "areally"

AUTHOR RESPONSE: Editorial suggestion taken (line 102).

27.    L 98: Sentence starting with "Nitrous oxide": the use of parenthesis seems odd.

AUTHOR RESPONSE: Please see point 7.

28.    L 104: Check the sentence – placement of "for" in the next line seems odd.

AUTHOR RESPONSE: The word "denitrification" had been accidentally omitted. The revised version of the text has now been re-written so this omission is no longer an issue.

29.    L 152: I like how the authors also analyzed topographic landforms. However, through- out the paper it is not clear, how these landforms were binned and weighted to form a habitat-wide data sets. Also, where were the samples taken from for the laboratory manipulations? Further, can the terminology be kept a bit more consistent? Throughout the manuscript, it is referred to as topography, landscape feature, landform, and basin landform. I assume they are all the same, but I suggest to use a consistent designation for this categorical variable.

AUTHOR RESPONSE: Topography/landform was treated as a categorical variable in our repeated measures ANOVA or ANCOVA tests. For the laboratory incubations, two soil cores were sampled from each landform. With respect to terminology; we have attempted to revise the text so that a narrower range of terminology is now employed.

30.    L250: This sentence essentially repeats the statement in L240

AUTHOR RESPONSE: This sentence has been removed in the revised manuscript.

31.    L260: I assume the amount of litter added corresponds to the amount of litter falling in 1 month?

AUTHOR RESPONSE: Yes.

32.    L483: Did you test for oxygen as a predictor, or was oxygen only assessed one time?

AUTHOR RESPONSE: Soil oxygen content was measured every time soil gas flux was sampled.

33.    L506: >24 hour incubation: Over what period were the fluxes averaged?

AUTHOR RESPONSE: The overall period for the incubation was 48 hours. For the late phase of the incubation, we calculated the flux rate over 24 to 48 hours. The text has now been revised to make this clearer (lines 546-549, 655-658).

34.    L667: Again, how long is the >24h period?

AUTHOR RESPONSE: Please see point 31.

35.    L726: The figure shows that premontane habitat is significantly different from the other, and not that the lower elevation forests (premontane, and lower montane forest) are significantly different from the higher elevation forests.

AUTHOR RESPONSE: The text has been corrected (see point 22).

36.    L835: check the sentence starting with "Moreover,. . ."

AUTHOR RESPONSE: This section has been revised; see point 10.

37.    L859: This sentence is not clear. What do the authors mean by "This pattern"

AUTHOR RESPONSE: We were referring to the overall trend of decreasing N2O flux with increasing elevation. The sentence has been removed in the revised version of the text.

38.    L884: It is hard to believe that NO3 additions did not stimulate N2O emission. Just eye-balling Fig 5 suggests, it seems that N2O flux over the incubation period increased with increasing NO3 levels added. Is there some artifact because of the way the ANOVA has been done (admittedly this is a weak point on my part – but maybe a recheck and some explanation is possible to enlighten me and the readers)?

AUTHOR RESPONSE: When evaluating for the effect of N addition level on N2O flux, the ANOVA pooled data across all other categories (i.e. site, incubation phase) to compare the difference in N2O flux among N treatments. Because of the high level of variability in N2O flux among study sites and incubation phases, the net effect was that the ANOVA found no clear signal of N addition level alone. The lack of trend is not an artefact of the ANOVA calculation per se, but rather represents the high level of variability among soils from different study sites and differing responses of N2O flux during different incubation phases.

39.    Supplementary figure: Please add the habitat to the x-axis for completion

AUTHOR RESPONSE: Editorial suggestion taken.

RESPONSE TO REFEREE 3

40.    Diem et al. report on a remarkably large and comprehensive set of observations and experiments examining N2O fluxes across the Kosnipata tropical elevation gradient in Peru. This was clearly a lot of work. The combination of high temporal resolution chamber observations with WFPS, 15N and litter experiments makes the study particularly compelling. I have four suggestions. First, there a few aspects of the 15N tracer work that require further clarification. Second, I recommend the authors consider scaling their observations to annual values. Third, depending on details of the 15N tracer methods, I suggest the authors consider making use of the N2: N2O flux ratios from the incubations to estimate total N gas losses from these ecosystems if appropriate. Finally, I think the authors could do a better job at contextualizing their work with reference to other studies and its global implications.

AUTHOR RESPONSE: We thank the referee for the positive remarks on our manuscript and constructive suggestions provided below.

41.    15N tracers: It would appear that the WFPS experiment was not a true "tracer" experiment but is also a N addition experiment and is therefore confounded. For the lower elevation sites, 200 ug N/g soil is not trivial. Are you sure that the background NO3 values are correct? The reported NO3-N values from soil extractions of ~150 ug/g are approximately 5-10 times higher than those observed in across most high N old- growth tropical forests worldwide. Tracer experiments often add < 0.5 ug/g at 15NO3 of ~99 atom percent. Further, unless I missed it, there is no description of the isotopic enrichment levels (per mil or atom percent). This needs to be included.

AUTHOR RESPONSE: Upon closer inspection, we realised that the values reported in the table are incorrect, and that the actual amount of N added was in fact much smaller than reported in the text. For example, for the WFPS experiment, the added amounts were 200 ng N/g soil for the lower elevation sites and 20 ng N/g soil for the higher elevation ones. For the N addition experiment, the values of N reported are the total amount of N added for the soil sample, and need to be normalised so that the values are reported on a per g soil basis. Thus, the true amounts of 15N tracer added in both the WFPS and N addition experiments are in fact in-line with the "trace" amount more typical of these types of 15N labelling experiments. This has been now corrected in the revised version of the text (Table 2). With respect to the level of isotopic enrichment; we applied the tracers at a 30 atom % level (see lines 272 and 350).

42.    Scaling: Given the seasonal representation of the sampling, I think annual scaling could be justified. When scaled annually, the mean N2O-N emissions (0.27 mg N m-2 day-1) would be ~ 0.98 kg N ha-1 yr-1 with peak fluxes of ~2.7 kg N ha-1 yr-1. On average, chamber studies and models find that N2O losses from undisturbed humid tropical soils are ~1-4 kg N ha-1 yr-1 (See van Lent et al. Biogeosciences 2015 and Werner et al. Global Biogeochemical Cycles 2007). So, these values fit right in.

AUTHOR RESPONSE: In-line with referee' suggestions, we have now produced simple area-and seasonally-weighted annual flux estimates in the Discussion of the revised text (section 6.3, Table 4).

43.    N2 fluxes: Given the response to the first point above, I suggest considering approximating total N gas losses from these ecosystems. Despite potential artifactual contributions of the incubations (disturbance, N additions) one could calculate rough N2 losses assuming equal N2:N2O ratios at a given WFPS as measured during the chamber work. This could be insightful as there are many chamber-based N2O estimates for tropical forests published but very few for total N gas fluxes because it's difficult to measure. Eyeballing the 15N2 versus 15N2O flux ratios (~20 to 80) and applying these to the chamber observations would yield N2 fluxes of ~20 – 216 kg N ha-1 yr-1. The lower-end flux is possible (see Fang et al. PNAS 2015) but the upper end estimate is highly unlikely. Such total N export rates could never persist in a near-equilibrium forest as even the lower end is higher than average N mineralization and annual plant uptake and far exceeds external N inputs in tropical forests (see Brookshire et al. Geophysical Research Letters 2017).

AUTHOR RESPONSE: In-line with the referee's suggestions, we have now revised the Discussion to incorporate estimates of N2 flux and gaseous N export (section 6.3, Table 4).

44.    The beauty of the Kosnipata gradient is that it represents a quasi-space-for-climate change substitution. More could be done with this context in the introduction and discussion. Further there are many other papers examining denitrification in tropical landscapes (some of them mentioned here) that would benefit the narrative to include.

AUTHOR RESPONSE: This remark is in agreement with concerns raised by other referees, and the Introduction and Discussion have been revised accordingly.

RESPONSE TO REFEREE 4

45.    Diem et al. present a comprehensive set of lab and field data relating to controls of soil nitrous oxide flux across an elevation gradient in the Peruvian Andes. As both long-term field measurements and lab-based manipulations are included, they are able to approach the discussion of N2O fluxes in these ecosystems from several different directions. This was excellent work that will be a valuable addition to our current knowledge of N-oxide fluxes and tropical montane ecosystems. However, the authors could really improve the paper by taking some additional time to craft a more integrated presentation/summation of their study. The results section, in particular, should be revised. A well-designed table or figure (or combination) could provide a fascinating and useful summation of the different experiments, while eliminating the repetitive text. Instead, the text of the results section should highlight the most important results – much of this could be moved from the discussion section, which can then be condensed and re-focused to provide a bit more literature context about the different aspects of the results being discussed.

AUTHOR RESPONSE: We thank the referee for the positive remarks on our manuscript and constructive suggestions provided below.

46.    Line 105: substrates for _____?

AUTHOR RESPONSE: This error has now been corrected in the revised version of the text. Please see point 24.

47.    Line 138: give average temperature range over the course of the study

AUTHOR RESPONSE: Mean annual temperature is provided in Table 1.

48.     Line 161: change 'because of' to 'due to'

AUTHOR RESPONSE: Editorial suggestion taken.

49.     Line 172: provide volume of chamber

AUTHOR RESPONSE: The chamber volume was approximately $0.008m^3$ (8 L); the text has been revised accordingly (line 200).

50.     Line 179: specify intervals

AUTHOR RESPONSE: Gas samples were collected at evenly spaced intervals over a 30 minute period; i.e. samples were collected 7.5 minutes apart. The text has been revised accordingly (line 206).

51.     Line 187-192: were zeroes included?

AUTHOR RESPONSE: Yes. The text has been revised accordingly (lines 219-220).

52.     Line 227-230: provide more detail: soil samples were taken in the field, air-dried and then re-wetted to target WFPS?

AUTHOR RESPONSE: To clarify, the WFPS experiments were conducted with field-moist samples; i.e. the soil samples were collected from the field, shipped to Aberdeen, and subsequently distributed into glass jars without being fully air-dried. For incubations where the target WFPS was below the field moisture levels, the soils were allowed to partially air-dry until they reached a value 10 % below the target WFPS for the experiment, and then carefully re-wetted through the 15N tracer application to bring up the soil moisture up to the target levels. For treatments where the target WFPS was above field moisture levels, the soils were simply wetted to 10 % below the target WFPS and then the 15N tracer solution added to bring the soil up to the target moisture level. The text has been revised accordingly (lines 258-262).

53.     Line 231-233: needs clarification: 0-10 cm depth included the organic layer at all elevations, except in the upper montane forest where 0-10 cm depth included only mineral? If 0-10 sometimes included the organic layer, what was the thickness of the organic layer at those elevations? What was the thickness of the organic layer at the upper montane site; how deep did you go to access the 0-10 mineral sample? explain reasoning behind this sampling decision; could this have affected your results?

AUTHOR RESPONSE: For premontane forest, lower montane forest, and montane grassland, the organic matter in the upper 10 cm soil layer is intermixed with the mineral phase, and does not constitute a distinct mineral-free horizon. Thus, we simply sampled from the 0-10 cm depth because there was no practical means of separating the organic matter from the mineral soil in these habitats. In contrast, upper montane forest soil shows a very different pattern of vertical stratification compared to the other habitats. In this habitat, the mineral soil is overlain by a thick (up to 17 cm deep) mineral-free organic layer, consisting of poorly decomposed leaves, roots, and humic materials; very akin to low density peat. To sample the mineral soil in this habitat, we went below this distinct organic horizon to a depth of approximately 17 cm. This information has now been added to the revised manuscript (lines 262-270).

With respect to the WFPS experiment; we decided to collect mineral soil from below the organic horizon in the upper montane forest because there was no mineral material found in this layer, making it difficult to compare results between habitats (given that the other habitats contain mineral material in the upper 10 cm of their soil profiles). At the time, we did not consider sampling the organic layer as well. This was an oversight on our part, which we tried to partially correct in our N addition experiments, by including the organic layer in those subsequent experiments.

54.    Line 297-307: clearly distinguish between 'soil core' and 'soil sample'; "core" implies that the soil is still intact – once it has been mixed and added to the jars, the soil samples are no longer soil cores

AUTHOR RESPONSE: Editorial suggestion taken (lines 336-348).

55.    Line 300-301: unclear; the five cores were mixed and then split into four equal parts? was the subsample and WFPS adjustment done on the cores or on the mixed soil in the jars?

AUTHOR RESPONSE: Each of the cores was split into four equal parts. The text has been revised to clarify this point (lines 341-342).

56.    Line 375: change 'with' to 'and'

AUTHOR RESPONSE: Editorial suggestion taken.

57.    Line 462: followed by topography

AUTHOR RESPONSE: Editorial suggestion taken.

58.    Line 473: change 'is' to 'was'

AUTHOR RESPONSE: Editorial suggestion taken.

59.    Line 474: define the fluctuation or refer to a table or figure where it is defined

AUTHOR RESPONSE: Editorial suggestion taken.

60. Line 585: change 'for' to 'from'

AUTHOR RESPONSE: Editorial suggestion taken.

61. Line 761: change semicolon to comma

AUTHOR RESPONSE: Editorial suggestion taken.

62. Line 768: between soil temperature and ___?

AUTHOR RESPONSE: N2O; text has now been corrected.

63. Line 779: change 'as' to 'at'

AUTHOR RESPONSE: Editorial suggestion taken.

64. Line 782: change 'are' to 'is'

AUTHOR RESPONSE: Editorial suggestion taken.

65. Line 836: remove 'and'

AUTHOR RESPONSE: Editorial suggestion taken.

[revised manuscript text omitted]

Page 43: [10] Deleted          Teh, Yit Arn          22/07/2017 20:05:00

---

## Author Response (AR2)

Dear Prof Joos

Once more, we would like to thank you and the referees for their efforts in reviewing our manuscript and providing us with the opportunity to further improve it. We have provided a brief response to the second referee's remaining comments below, and hope that the changes we have made in response to these suggestions meet with your approval. We include a copy of the manuscript with track changes enabled to allow you to more quickly identify where we have made modification to the text.

Yours sincerely,
Yit Arn Teh

RESPONSE TO REFEREE 2
1. Thank you so much to the authors for taking all the reviewers' suggestion to heart. I do like the manuscript, and the improvement are satisfactory. This is a very important manuscript, where the work neatly combines field measurements and laboratory experiments to tease out drivers of nitrous oxide emissions. A few comments remain, but they probably more reflect my ignorance, and elsewhere are more of editorial nature.

AUTHOR RESPONSE: Our sincerest thanks for these kind remarks. We hope the responses we have provided below will be to the referee's satisfaction.

2. Abstract: I am not sure whether unweighted means need to be part of the abstract, as the weighting really more reflect the montane tropical N2O emissions.

AUTHOR RESPONSE: Because our area- and seasonally-weighted annual flux estimates were conducted as part of a post-hoc exercise, and may contain errors associated with calculations of the areal fractions or may have errors associated with our simple method of temporal weighting, we believe it may be more prudent to indicate both unweighted and weighted annual flux estimates in the abstract. We have since updated the current abstract to include both unweighted and weighted flux estimates (see L35-L41).

3. Abstract L35: Can it be added that this conclusion stems from 15N enrichement experiment?

AUTHOR RESPONSE: We apologise if we have misunderstood the referee, but the data reported in L35 refer to field fluxes. The 15N-nitrate addition experiment did verify that premontane forest showed significantly higher 15N-N2O production potential than montane grassland and upper montane forest mineral layer soil (see L682-L688), but we were not referring to these data here.

4. Abstract L 53: I suggest to start the last sentence with "Yet, nitrous oxide flux..." to highlight the contrast to the finding of N2O emission being a function of N richness.

AUTHOR RESPONSE: Editorial suggestion taken.

5. L98 suggest to use "large" instead of "long"

AUTHOR RESPONSE: Editorial suggestion taken. We have also corrected the title (L3) and other parts of the text (L23) to remain internally consistent.

6. L115 This seems at odds with the data, for most of the time WFPS is < 60%

AUTHOR RESPONSE: We must apologise for the lack of precision in our language. Here we were referring to our preliminary dataset, where there were a large proportion of values that were near to or above the 60 % WFPS threshold. The referee is correct that in the longer-term dataset presented in this manuscript, WFPS was generally lower than this threshold, which suggests that the preliminary data were collected during a year that was slightly wetter than the overall average for the whole study period. We have now revised the text to make it clearer that what we are referring to here is our preliminary dataset (L129-L130).

7. Paragraph starting with L 121: In my previous review I suggested to try to work with lags. The authors countered that the time series is too short for doing that. While I am not sure that 30 months is long enough for lag analysis, the visual lag correlation is striking. At the same time, it is probably not the reviewer's choice on how the analysis should be conducted.

AUTHOR RESPONSE: The referee is correct that there do appear to be lags in the data, but we were unable to pick this out with the statistical analyses that we used. We suspect that these lags may have been obscured by the large variance in the field data, making it difficult to identify statistically significant trends.

8. L233: Check throughout the manuscript to use consistently use "litter-fall" or "litterfall".

AUTHOR RESPONSE: Editorial suggestion. We have corrected the manuscript so that we use the word "litter-fall" consistently throughout the text. The word "litterfall" was only used once, whereas we "litter-fall" was used in all other instances throughout the text, so we thought it more consistent to use the former form of the word.

9. L241: Not clear what "at the bottom" means when you give a depth range.

AUTHOR RESPONSE: We have now changed the phrase to "deployed in the plant rooting zone" to make the sentence more economical.

10. L261: Can you indicate the N concentration (eg. ug/L), of the isotope solution added?

AUTHOR RESPONSE: The N concentration of the 30 atom % 15N tracer varied depending on the habitat (Table 2), so we chose to keep the explanation simpler in this part of the text so as not to make the methods more difficult to read. We would prefer to keep the text as it is, for the sake of simplicity of explanation.

11. Result section: I really appreciate the extra work the authors put in to make this section much more readable.

AUTHOR RESPONSE: Thank you for this comment. We really valued this referee's comments and those of the other referees on the Results, because their suggestion have improved the readability of the text.

12. L 817: Does this contradict  L 800, where you say that N poor habitats have a greater proportional response (where you suggest that there is a response to NO3 addition)?

AUTHOR RESPONSE: Thank you for this comment. One again, the referee has highlighted where we should have been more precise with our language. The text has been revised to better-represent what we were trying to express (L833-L851). In this part of the text, we were trying to explain why we may not have observed an immediate response of the soil microbial community to enhanced NO3- availability.

13. Figure 3: replace "broken line" with "dashed line"

AUTHOR RESPONSE: Editorial suggestion taken.

[revised manuscript text omitted]
 | 0.79 ± 0.26 a | 51.9 ± 1.6 a | 51.2 ± 2.1 a | 20.7 ± 0.1 a | 20.2 ± 0.1 b | 21.5 ± 0.3 | 20.4 ± 0.5 | 19.4 ± 0.2 a | 19.6 ± 0.2 a | 23.2 ± 3.6 a | 22.1 ± 2.1 a | 31.4 ± 13.0 | 11.3 ± 1.8 |
| | n = 130 | n = 98 | n = 135 | n = 135 | n = 143 | n = 120 | n = 143 | n = 120 | n = 52 | n = 36 | n = 89 | n = 96 | n = 90 | n = 95 |
| Lower montane | 0.09 ± 0.08 a | 1.02 ± 0.58 b | 42.2 ± 1.0 a | 34.0 ± 1.4 b | 18.1 ± 0.1 a | 17.3 ± 0.2 b | 18.9 ± 0.3 | 18.3 ± 0.2 | 19.2 ± 0.2 a | 19.2 ± 0.1 a | 11.8 ± 1.9 a | 7.8 ± 1.4 a | 20.2 ± 5.4 | 8.6 ± 0.9 |
| | n = 212 | n = 137 | n = 271 | n = 179 | n = 254 | n = 164 | n = 254 | n = 164 | n = 146 | n = 81 | n = 123 | n = 94 | n = 124 | n = 93 |
| Upper montane | 0.06 ± 0.09 a | 0.01 ± 0.11 a | 42.0 ± 1.3 a | 24.3 ± 1.4 b | 11.8 ± 0.1 a | 10.9 ± 0.2 b | 12.8 ± 0.2 | 12.5 ± 0.3 | 18.7 ± 0.2 a | 18.5 ± 0.2 a | 1.4 ± 0.2 a | 0.6 ± 0.2 b | 22.5 ± 6.3 | 11.3 ± 1.4 |
| | n = 207 | n = 146 | n = 264 | n = 180 | n = 255 | n = 165 | n = 255 | n = 165 | n = 165 | n = 109 | n = 128 | n = 91 | n = 129 | n = 93 |
| Montane grassland | -0.01 ± 0.11 a | 0.19 ± 0.12 a | 88.5 ± 0.3 a | 88.3 ± 0.5 a | 11.6 ± 0.1 a | 9.0 ± 0.2 b | 11.4 ± 0.3 | 12.0 ± 0.5 | 12.2 ± 0.9 a | 15.4 ± 0.8 b | 1.5 ± 0.4 a | 2.1 ± 0.4 a | 17.8 ± 4.3 | 7.2 ± 0.8 |
| | n = 238 | n = 160 | n = 303 | n = 184 | n = 282 | n = 205 | n = 284 | n = 205 | n = 176 | n = 117 | n = 128 | n = 81 | n = 135 | n = 84 |

Table 4. Area- and seasonally-weighted annual estimates of $N_2O$, $N_2$, and total gaseous N flux

| Elevation Band (m.a.s.l.) | Habitat | Surface Area (ha) | Fraction of Land Area | Fraction of Year Wet Season | Fraction of Year Dry Season | Nitrous Oxide Yield | Unweighted Nitrous Oxide Flux Wet Season kg $N_2O$-N ha$^{-1}$ yr$^{-1}$ | Unweighted Nitrous Oxide Flux Dry Season kg $N_2O$-N ha$^{-1}$ yr$^{-1}$ | Area-weighted Nitrous Oxide Flux Wet Season kg $N_2O$-N ha$^{-1}$ yr$^{-1}$ | Area-weighted Nitrous Oxide Flux Dry Season kg $N_2O$-N ha$^{-1}$ yr$^{-1}$ | Area-weighted and Seasonally-weighted Annual Estimate of $N_2O$ Flux kg $N_2O$-N ha$^{-1}$ yr$^{-1}$ | Area-weighted and Seasonally-weighted Annual Estimate of $N_2$ Flux kg $N_2$-N ha$^{-1}$ yr$^{-1}$ | Area-weighted and Seasonally-weighted Annual Estimate of Total Gaseous N Flux kg N ha$^{-1}$ yr$^{-1}$ |
|---|---|---|---|---|---|---|---|---|---|---|---|---|---|
| 600-1200 | Premontane forest | 733000 | 0.24 | 0.58 | 0.42 | 0.4 ± 0.05 | 2.59 ± 0.91 | 2.88 ± 0.95 | 0.63 ± 0.22 | 0.70 ± 0.23 | 0.66 ± 0.16 | 1.00 ± 0.29 | 1.66 ± 0.33 |
| 1200-2200 | Lower montane forest | 892000 | 0.30 | 0.58 | 0.42 | 0.19 ± 0.04 | 0.33 ± 0.29 | 3.72 ± 2.12 | 0.10 ± 0.09 | 1.10 ± 0.63 | 0.52 ± 0.27 | 2.21 ± 1.24 | 2.73 ± 1.26 |
| 2200-3200 | Upper montane forest | 807000 | 0.27 | 0.58 | 0.42 | 0.42 ± 0.05 | 0.22 ± 0.33 | 0.04 ± 0.40 | 0.06 ± 0.09 | 0.01 ± 0.11 | 0.04 ± 0.07 | 0.05 ± 0.09 | 0.09 ± 0.12 |
| 3200-3700 | Montane grasslands | 586000 | 0.19 | 0.58 | 0.42 | 0.61 ± 0.06 | -0.04 ± 0.40 | 0.69 ± 0.44 | -0.01 ± 0.08 | 0.13 ± 0.09 | 0.05 ± 0.06 | 0.03 ± 0.04 | 0.09 ± 0.07 |
| Totals | | 3020000 | | | | | | | | | 1.27 ± 0.33 | 3.29 ± 1.27 | 4.57 ± 1.31 |

**Figure 1.** Map of study sites across the Kosñipata Valley, Manu National Park, Peru.

[Figure]

**Figure 2.** Plot-averaged (a) net $N_2O$ flux, (b) water-filled pore space, and (c) resin-extractable

$NO_3^-$ flux among habitats. Boxes enclose the interquartile range, whiskers indicate the 90th and 10th percentiles. Lower case letters indicate statistically significant differences among means (Fisher's LSD, $P < 0.05$).

[Figure]

**Figure 3.** Time series of net N$_2$O flux and water-filled pore space (WFPS). Panels indicate data for (a) premontane forest, (b) lower montane forest, (c) upper montane forest, and (d)

montane grasslands for the 30-month study period beginning in January 2011 and ending in

June 2013. The broken horizontal line running across each panel denotes the overall mean

N$_2$O flux or WFPS for that habitat. The dashed line in each box indicate median values and the black lines indicate means. Dry and wet seasons are denoted by vertical shading on the graph, with the dry season (May to September) highlighted in white and the wet season (October to April) in light blue.

[Figure]

**Figure 4.** Total (a) $^{15}N-N_2O$ flux and (b) $^{15}N-N_2$ flux during the early (≤24 hours) and late (>24

hours) incubation phases of the water-filled pore space (WFPS) experiment. Results from the

90 % WFPS treatment are shown in dark-grey, while data from the 70 %, 50 %, and 30 %

WFPS treatments are shown in mid-grey, light-grey, and white, respectively. The bar charts show means and standard errors.

[Figure]

 **Figure 5.** (a) $^{15}N$-$N_2O$ flux and (b) $^{15}N$-$N_2$ flux during the early (≤24 hours) and late (>24 hours)

incubation phases of the $NO_3^-$ addition experiment. Results from the +50 % $NO_3^-$ addition are shown in dark-grey, while data from the +100 % and +150 % treatments are shown in mid- grey and light-grey, respectively. The bar charts show means and standard errors.